# SCENE: Multi-Robot Active Cognition via Unified Free-Energy Minimization

## ABSTRACT

Existing multi-robot exploration methods are fragmented, leading to suboptimal decisions in complex environments. We introduce SCENE, a framework that unifies decision-making by reframing exploration as cognitive free-energy minimization Wakayama & Ahmed (2023). Our core contribution is a free-energy formulation that integrates geometric uncertainty, semantic novelty, and the Age of Information (AoI). The system leverages a neural implicit field for unified representation, a prototypical network for open-world semantics Yang et al. (2024), and a Graph Neural Network (GNN) for emergent collaboration. Trained end-to-end via self-supervision, SCENE surpasses state-of-the-art methods in challenging scenarios, demonstrating superior efficiency and decision-making intelligence.

## 1 INTRODUCTION AND RELATED WORK

Prevailing multi-robot exploration methods are often limited by treating geometric, semantic, and dynamic objectives as decoupled problems. This fragmentation leads to suboptimal trade-offs, as agents lack a unified principle for decision-making, such as choosing between exploring a new area versus inspecting a novel object. We propose a principled alternative to such isolated heuristics: framing exploration as a process of **cognitive free-energy minimization** Haddon-Hill & Murata (2024). This principle posits that intelligent action is driven by minimizing the discrepancy between an agent's internal model and its perception of external reality. We instantiate this principle in **SCENE**, a framework founded on a unified free-energy objective. This single objective function intrinsically balances **geometric uncertainty, semantic novelty, and information timeliness (Age of Information)** Kaul et al. (2012), providing a structured solution to the decision-making trade-off. Through an end-to-end, self-supervised system, we demonstrate that this approach facilitates emergent collaboration and achieves superior performance, shifting the objective from pure autonomous mapping towards comprehensive autonomous cognition. Current multi-robot exploration methods are fragmented. Classical approaches, such as frontier-based (**NBVP-FKIE**) Naazare et al. (2022) or entropy-based (**EGE**) Carrillo et al. (2015) strategies, are confined to geometric objectives. While recent learning-based methods like **SEAL** Chaplot et al. (2021) introduce semantic awareness, they lack a principled framework to unify semantic goals with mapping. Furthermore, most systems are designed for static environments, failing to reason about information timeliness (Age of Information) Kaul et al. (2012), a feature none of our evaluated baselines possess. The rise of neural implicit representations (e.g., **NeuV-SLAM**) Guo et al. (2025) has revolutionized passive mapping. However, their use in active perception remains nascent, typically involving simple, non-unified planners. SCENE bridges these gaps by leveraging a neural field as the backbone for a unified free-energy objective that intrinsically balances geometric, semantic, and dynamic information gain for intelligent, collaborative decision-making.

## 2 SCENE FRAMEWORK OVERVIEW AND PIPELINE

The SCENE framework is a closed-loop system that tightly couples **World Modeling**, **Collaborative Decision-Making**, and **Policy Learning** through a unified free-energy objective. The pipeline operates in continuous cycles as follows:

**Step 1: Building the Unified World Representation.** The team collaboratively constructs a shared world model using a **hash-encoded multi-head neural implicit field**. This function, $\Phi_\theta(x, t)$, has

three specialized output heads for geometry, semantics, and dynamics, and is trained online and self-supervisedly by all robots.

**Step 2: Quantifying the Cognitive State.** We quantify the team's understanding of the world via the **cognitive free energy,** $\mathcal{F}$. Its three key components are computed from the world model: Global Geometric Uncertainty, Global Semantic Novelty (using a prototype-based memory), and Global Age of Information (from a dedicated AoI neural field).

**Step 3: Collaborative Action Generation and Evaluation.** Each robot generates candidate actions, encodes the top $K$ as **action intents**, and broadcasts them. A **Graph Neural Network (GNN)** then processes these intents, allowing each robot to reason about team-level synergy and select an action that maximizes the expected reduction in the team's total free energy.

**Step 4: Execution and Learning.** The chosen action is executed. The system is trained end-to-end via Multi-Agent Reinforcement Learning, where the team's **shared reward signal** is the **actual measured decrease in global free energy** $\mathcal{F}$. This directly incentivizes the policy to learn efficient, collaborative exploration behaviors.

# 3 SCENE: CORE ARCHITECTURE AND ALGORITHM DESIGN

Our methodology addresses a central question: how can a multi-robot team learn an efficient, collaborative policy to actively and comprehensively cognize a dynamic, semantic-rich world under a unified, theoretically sound framework? Our design follows a logical hierarchy from defining the objective, to constructing the representation, to learning the solution.

## 3.1 EXPLORATION AS COGNITIVE FREE-ENERGY MINIMIZATION

We redefine exploration as **cognitive free-energy minimization** Haddon-Hill & Murata (2024), an approach rooted in computational neuroscience Da Costa et al. (2022). This principle posits that an agent acts to minimize the discrepancy between its internal world model and sensory evidence, which is equivalent to maximizing the Expected Information Gain (EIG) about the world state $M$ Zhou et al. (2025). The objective is to select a collective action $a$ that maximizes the EIG, which we formulate as minimizing the cognitive free energy $\mathcal{F}(a) = -\text{EIG}(a)$.

The core innovation of SCENE is defining the world state $M$ as a unified cognitive state encompassing **Geometry (G)**, **Semantics (S)**, and **Dynamic Timeliness (D)**. This allows us to decompose the total EIG into a weighted sum of three semantically orthogonal components, forming the decision-making core of the framework:

$$\text{EIG}(a) \approx w_g \cdot \text{EIG}_G(a) + w_s \cdot \text{EIG}_S(a) + w_d \cdot \text{EIG}_D(a) \tag{1}$$

where $w_g, w_s, w_d$ are task-dependent weighting factors that modulate priorities. Each component represents a distinct cognitive drive:

1. **Geometric Information Gain ($\text{EIG}_G$):** Measures the expected reduction in *geometric uncertainty*. It drives the classic impulse to map unknown areas, resolving questions about where surfaces exist versus where free space lies.
2. **Semantic Information Gain ($\text{EIG}_S$):** Measures the expected *semantic surprise* or novelty. This drives the robot to discover and scrutinize meaningful, novel entities that are improbable under the current semantic model.
3. **Dynamic Information Gain ($\text{EIG}_D$):** Measures the expected contribution to maintaining *knowledge freshness*. It quantifies the value of reducing uncertainty caused by the passage of time (high Age of Information), crucial for situational awareness in non-stationary environments.

To operationalize this principle, we formalize the EIG for a candidate action $a$ as an integral over its potential field of view, $\mathcal{V}_a$. The total gain is the expected sum of information densities at each spatial location $\mathbf{x} \in \mathcal{V}_a$, weighted by the visibility $V(\mathbf{x}|a)$:

$$\text{EIG}(a) = \int_{\mathbf{x} \in \mathcal{V}_a} (w_g \mathcal{I}_G(\mathbf{x}) + w_s \mathcal{I}_S(\mathbf{x}) + w_d \mathcal{I}_D(\mathbf{x})) \, V(\mathbf{x}|a) \, d\mathbf{x} \tag{2}$$

Here, $\mathcal{I}_G(\mathbf{x})$, $\mathcal{I}_S(\mathbf{x})$, and $\mathcal{I}_D(\mathbf{x})$ represent the geometric uncertainty, semantic novelty, and dynamic timeliness information densities at location $\mathbf{x}$, respectively. By framing the problem as a descent

on this unified free-energy landscape, the complex trade-offs in exploration are resolved via a single, principled objective.The following section details our unified world representation, which is specifically designed to serve as the computational backbone for instantiating and evaluating these information density terms. A more detailed theoretical derivation of the Expected Information Gain (EIG) decomposition is provided in Appendix A.

### 3.2 DECISION-ORIENTED UNIFIED WORLD REPRESENTATION

To compute the unified information gain defined in Eq. equation 1, we require a world representation that is continuous, differentiable, and capable of jointly encoding geometric, semantic, and dynamic information. To this end, we design a **hash-encoded multi-head neural implicit field,** $\Phi_\theta$, inspired by the efficiency of Instant Neural Graphics Primitives (Instant-NGP) Müller et al. (2022). This representation serves as the computational backbone for evaluating our free-energy objective Zhu et al. (2022).

#### 3.2.1 ARCHITECTURE

The field $\Phi_\theta$ maps a multi-resolution hash-encoded spatio-temporal coordinate $\gamma(x, t)$ to three distinct, task-specific outputs:

$$\Phi_\theta : \gamma(x, t) \mapsto \{h_g, h_s, h_d\} \tag{3}$$

where $x \in \mathbb{R}^3$ is a spatial location and $t$ is the timestamp. Each head is designed to compute a specific component of the information density in Eq. equation 2.

- **Geometric Head** ($h_g$): This head is a Multi-Layer Perceptron (MLP) that outputs the Truncated Signed Distance Function (TSDF) value $d(x, t)$ and a feature vector $\mathbf{f}_g(x, t)$ Qi et al. (2022). To quantify geometric uncertainty, we maintain a running variance $\sigma_g^2(x, t)$ of the TSDF predictions from multiple viewpoints. The geometric information density $\mathcal{I}_G(x)$ is then directly defined as this variance, which drives agents to explore regions with high structural ambiguity:

$$\mathcal{I}_G(\mathbf{x}, t) = \sigma_g^2(\mathbf{x}, t) \tag{4}$$

- **Semantic Head** ($h_s$): To robustly estimate semantic category distributions and, critically, the model's *epistemic uncertainty*, we employ a **Deep Ensemble** of $N_e$ independent semantic heads Lakshminarayanan et al. (2016), $\{h_s^j\}_{j=1}^{N_e}$. Each head takes the geometric feature vector $\mathbf{f}_g$ as input and outputs a logit vector $\mathbf{z}_j(x, t)$ over $K$ known semantic classes. The epistemic uncertainty, which captures what the model does not know about the known classes, is approximated by the mutual information between the model parameters and the prediction. This is efficiently calculated from the ensemble's outputs:

$$\mathcal{I}_S^{\text{epistemic}}(\mathbf{x}, t) \approx H\left(\frac{1}{N_e} \sum_{j=1}^{N_e} p_j(\mathbf{x}, t)\right) - \frac{1}{N_e} \sum_{j=1}^{N_e} H(p_j(\mathbf{x}, t)) \tag{5}$$

where $p_j(\mathbf{x}, t) = \text{Softmax}(\mathbf{z}_j(\mathbf{x}, t))$ and $H(\cdot)$ is the entropy. This term drives agents to regions where the ensemble members disagree.

- **Dynamic Head** ($h_d$): This head is a small MLP that predicts a scalar value $v(x, t) \in [0, 1]$, representing the probability of a state change (geometric or semantic) at location $x$ in the near future. This prediction is supervised by temporal inconsistencies in observations and provides a forward-looking estimate of where the world model is likely to become obsolete.

#### 3.2.2 AUXILIARY COGNITIVE MODULES

Two auxiliary modules support the complete calculation of the information gain, particularly for open-world novelty and timeliness.

- **Open-World Semantic Memory:** To compute novelty beyond uncertainty over $K$ known classes, we adopt a **Prototypical Network** approach Snell et al. (2017). Each robot maintains a memory of semantic prototypes, $\{\mathbf{p}_k, \boldsymbol{\Sigma}_k\}_{k=1}^K$, representing the mean and covariance of feature vectors for each class. When a new observation yields a semantic feature $\mathbf{f}_s(x, t)$, its novelty is quantified by the Mahalanobis distance to the closest prototype, which approximates the negative log-likelihood under a Gaussian Mixture Model. This

forms the novelty component of semantic information:

$$\mathcal{I}_S^{\text{novelty}}(\mathbf{x}, t) = \min_k \left( (\mathbf{f}_s(\mathbf{x}, t) - \mathbf{p}_k)^T \boldsymbol{\Sigma}_k^{-1} (\mathbf{f}_s(\mathbf{x}, t) - \mathbf{p}_k) \right) \tag{6}$$

The total semantic information density is a weighted sum of the epistemic and novelty components, $\mathcal{I}_S = \lambda_e \mathcal{I}_S^{\text{epistemic}} + \lambda_n \mathcal{I}_S^{\text{novelty}}$, driving discovery of both confusing known objects and genuinely new entities. The full mechanism for prototype initialization, online updates, and new category detection is detailed in Appendix B.3.

- **Age of Information (AoI) Neural Field:** To track AoI over a continuous space, we introduce a second, lightweight hash-encoded neural field, $\Psi_\xi$, which learns the continuous AoI function, $A(x, t)$. For any point $x$ within the current sensor frustum, its target AoI is $A(x, t) = 0$. The growth of AoI over time is learned implicitly. The dynamic information density $\mathcal{I}_D(\mathbf{x}, t)$ is thus simply the current predicted AoI at that location:

$$\mathcal{I}_D(\mathbf{x}, t) = A(\mathbf{x}, t) \tag{7}$$

This formulation elegantly quantifies the value of revisiting a location as the amount of time-induced uncertainty that will be resolved, providing the basis for computing $\text{EIG}_D$.

### 3.2.3 SELF-SUPERVISED TRAINING

The entire world representation, including $\Phi_\theta$ and $\Psi_\xi$, is trained online and self-supervisedly from the robots' own sensor streams, requiring no manual labeling Thomas et al. (2021). The geometric head is supervised by the depth map from RGB-D cameras. The semantic heads are supervised by pseudo-labels generated by a fixed, pre-trained 2D segmentation model (e.g., SAM or a foundation model) Kirillov et al. (2023), with view inconsistencies being naturally averaged out by the ensemble. The dynamic head is supervised by detecting temporal differences between consecutive observations of the same location. This end-to-end self-supervised pipeline ensures the system's autonomy and adaptability to novel environments. Further details on the data flow and the formulation of the self-supervised loss functions are provided in Appendix B.2.

## 3.3 LEARNING A COLLABORATIVE DECISION-MAKING POLICY

Given the unified world representation and the free-energy objective, the final challenge is to derive a tractable, decentralized policy that enables the robot team to collaboratively select actions that minimize the global free energy. We formulate this as a **Multi-Agent Reinforcement Learning (MARL)** problem Li et al. (2023) and learn the policy within a **Centralized Training, Decentralized Execution (CTDE)** paradigm Liu et al. (2022).

### 3.3.1 ATTENTION-BASED COORDINATION VIA INTENT BROADCASTING

To circumvent the prohibitive communication bandwidth required for sharing full belief states, we design an efficient coordination mechanism based on the exchange of low-dimensional **Action Intents** Blumenkamp et al. (2021). The process for each robot $i$ at each decision step is as follows:

1. **Individual Proposal Generation:** Each robot $i$ first samples a set of candidate actions $\{a_j\}$. For each action, it performs a lookahead simulation using a lightweight predictive model to estimate the *individual information gain*, $\text{EIG}_{ind}(a_j)$, based only on its local view of the shared world model $\Phi_\theta$. This estimate reflects the action's value if the robot were acting alone.

2. **Intent Encoding and Broadcasting:** The robot selects its top-$K$ candidate actions based on $\text{EIG}_{ind}$ and encodes them into a compact **intent vector**, $\mathcal{I}_i \in \mathbb{R}^{K \times D_{intent}}$. This vector, which contains the goal coordinates and expected individual gain for each of the $K$ intentions, is then broadcast to all neighboring robots within its communication range.

3. **Collaborative Refinement with GAT:** Each robot constructs a local computation graph where nodes represent itself and its neighbors. The feature vector for each node $j$, $\mathbf{h}_j$, is an embedding of its state $\mathbf{s}_j$ and its broadcasted intent vector $\mathcal{I}_j$. We employ a **Graph Attention Network (GAT)** Veličković et al. (2017) to process this graph. The GAT computes a normalized attention coefficient $\alpha_{ij}$ for each neighbor $j \in \mathcal{N}_i \cup \{i\}$, quantifying the relevance of robot $j$'s intentions to robot $i$:

$$\alpha_{ij} = \frac{\exp\left(\text{LeakyReLU}\left(\mathbf{a}^T[\mathbf{Wh}_i || \mathbf{Wh}_j]\right)\right)}{\sum_{k \in \mathcal{N}_i \cup \{i\}} \exp\left(\text{LeakyReLU}\left(\mathbf{a}^T[\mathbf{Wh}_i || \mathbf{Wh}_k]\right)\right)} \tag{8}$$

where $\mathbf{W}$ is a shared learnable linear transformation, $\mathbf{a}$ is a weight vector for the attention mechanism, and $||$ denotes concatenation. The GAT then outputs a **coordination context vector**, $\mathbf{c}_i$, by computing an attention-weighted aggregation of the neighbors' transformed features:

$$\mathbf{c}_i = \sigma \left( \sum_{j \in \mathcal{N}_i \cup \{i\}} \alpha_{ij} \mathbf{W} \mathbf{h}_j \right) \tag{9}$$

where $\sigma$ is a non-linear activation function. This vector $\mathbf{c}_i$ implicitly encodes the optimal adjustment to robot $i$'s plan in light of the team's collective intentions, providing the necessary context to shift from greedy individual optimization to globally-aware collaborative decision-making.

### 3.3.2 DISTRIBUTED Q-NETWORK AND TRAINING

We utilize the value decomposition network architecture, QMIX Rashid et al. (2018), to facilitate efficient centralized training.

- **Q-Network Architecture:** Each agent $i$ uses a local Q-network, $Q_i$, to evaluate its candidate actions. The network takes as input its local state observation $\mathbf{s}_i$, the candidate action $a_j$, and critically, the coordination context vector $\mathbf{c}_i$ generated by the GAT. It outputs the action's final utility, $Q_i(\mathbf{s}_i, a_j, \mathbf{c}_i)$. This architecture allows the final decision to be conditioned on both local information and distilled collaborative context.
- **Reward Function:** The key to ensuring the learned policy aligns with our theoretical objective is the reward signal. The team receives a single, shared reward, $R_{team}$, at each timestep, defined directly as the **actual, measured reduction in global cognitive free energy**:

$$R_{team}(t) = \mathcal{F}(t-1) - \mathcal{F}(t) = \Delta \text{EIG}_{total}(t) \tag{10}$$

  where $\mathcal{F}(t)$ is the global free energy computed from the state of the world model $\Phi_{\theta_t}$ at time $t$. This reward function creates a direct optimization pressure to learn policies that are maximally efficient at reducing global uncertainty, discovering novelty, and maintaining information freshness, perfectly aligning the reinforcement learning objective with our first-principle-based exploration goal.
- **Training and Execution:** During the centralized training phase in simulation, the global free energy is accessible for computing the reward, and gradients can be passed through the QMIX network to train all individual $Q_i$ networks and the GAT parameters end-to-end. Once trained, the system is deployed for decentralized execution. Each robot only needs to run its local GAT and Q-network forward passes. This requires only the broadcasting of low-dimensional intent vectors, making the approach scalable and practical for real-world deployment with limited communication. A detailed algorithmic pipeline, along with network architectures and training hyperparameters, can be found in Appendices B.5 and B.6.

## 4 EXPERIMENTAL EVALUATION

To rigorously validate the performance, robustness, and intelligence of the SCENE framework, we conducted a series of comprehensive experiments in high-fidelity simulations. Our evaluation is designed to answer three key questions: 1) Does SCENE achieve superior overall performance in complex, dynamic, and semantic-rich environments compared to state-of-the-art methods? 2) How robust and scalable is the framework under extreme conditions such as perceptual degradation and communication failure? 3) Can we provide interpretable evidence that SCENE's unified free-energy objective leads to more intelligent, emergent collaborative behaviors?

### 4.1 EXPERIMENTAL SETUP

#### 4.1.1 SIMULATION PLATFORM & SCENARIOS

All experiments were conducted in **NVIDIA Isaac Sim**, a robotics simulator built on the **NVIDIA Omniverse™** platform Zhou et al. (2023), ensuring high-fidelity physics and photorealistic render-

ing. We designed three challenging benchmark scenarios to systematically probe the limits of our framework:

- *Dynamic Urban Intersection (DUI-25)*: A large-scale urban environment featuring numerous dynamic vehicles and pedestrians, designed to test long-term situational awareness and dynamic tracking.
- *Multi-Agent Rescue Mission (MARM-25)*: An unstructured indoor/outdoor disaster site with complex 3D structures and key semantic targets (e.g., rescue mannequins), designed to evaluate task-driven 3D exploration and semantic search.
- *Subterranean Challenge Environment (BCE-25)*: A long, procedurally generated cave system with sparse textures and challenging low-light conditions, adapted from the DARPA SubT Challenge Tranzatto et al. (2022) to test robustness in perceptually-degraded environments.

### 4.1.2 ROBOT PLATFORM

Our heterogeneous robot team consists of **8 simulated robots**: 6 ground vehicles (UGVs) and 2 aerial vehicles (UAVs) Mittal et al. (2023). Each is equipped with a simulated LiDAR and an RGB-D camera. Communication with our framework is managed via a ROS 2 Bridge.

## 4.2 COMPARATIVE METHODS

We benchmark SCENE against a comprehensive set of baselines and four targeted ablation studies:

- **Classic Methods**: We implement two foundational approaches: **NBVP-FKIE** Naazare et al. (2022), a geometric frontier-based exploration method, and **EGE** Carrillo et al. (2015), an entropy-based graph exploration strategy representing classic multi-robot coordination.
- **Modern State-of-the-Art**: We compare against two strong, specialized SOTA methods: **LIMO** Graeter et al. (2018) coupled with a Next-Best-View planner, representing high-precision geometric SLAM, and **SEAL** Chaplot et al. (2021), a leading self-supervised semantic exploration framework.
- **Cutting-Edge Neural SLAM**: We include **NeuV-SLAM** Guo et al. (2025) with an information-gain planner, representing the forefront of neural implicit SLAM systems.
- **Ablation Study**: **SCENE-Geo** (only geometric uncertainty), **SCENE-NoAoI** (no Age of Information objective), **SCENE-NoSem** (no semantic novelty objective), and **SCENE-Decentralized** (no GNN-based coordination).

## 4.3 EVALUATION METRICS

We employ a multi-level metric system to provide a holistic assessment:

- **Core Performance Metrics**:
  - *Cognitive Dissonance (CD)*: A unified score measuring the final map's inconsistency with ground truth. It is formally defined as a weighted sum of normalized errors across geometric, semantic, and dynamic domains, creating a holistic measure of cognitive error. Lower is better.

$$CD = \lambda_g \cdot RMSE_{geom} + \lambda_s \cdot (1 - mIoU_{sem}) + \lambda_d \cdot (1 - MOTA_{dyn}) \quad (11)$$

    The $\lambda$ coefficients are task-dependent weights. To ensure a fair comparison, these weights were held constant for all methods within a given scenario. A detailed explanation of their selection is provided in Appendix C.2.2.
  - *Energy Efficiency (EE)*: A measure of the total information acquired per unit of energy consumed, proxied by the total distance traveled by the team. Higher is better.

$$EE = \frac{\mathcal{V}_{exp} + \beta \cdot N_{tgt}}{\sum_{i=1}^{N_{robots}} \mathcal{D}_i} \quad (12)$$

    where $\mathcal{V}_{exp}$ is the explored volume, $N_{tgt}$ is the number of semantic targets found, and $\mathcal{D}_i$ is the distance traveled. The weighting factor $\beta$ balances exploration against target discovery. This factor was also kept constant for all methods in the relevant scenarios, as further detailed in Appendix C.2.2.

Table 1: Comprehensive Performance and Ablation Study Comparison in DUI-25 and MARM-25 Scenarios. A '−' indicates a metric is not applicable for a given method or scenario (e.g., no explicit coordination module or no dynamic objects to track).

| Method | Scene | Cognitive Dissonance (CD) ↓ | Task Time (s) ↓ | Energy Effic. (EE) ↑ | Geom. RMSE (m) ↓ | Semantic mIoU ↑ | Dynamic MOTA ↑ | Comp. Load (GFLOPS) ↓ | Coord. Overhead (%) ↓ |
|---|---|---|---|---|---|---|---|---|---|
| *Performance in Dynamic Urban Intersection (DUI-25)* | | | | | | | | | |
| **SCENE (Ours)** | DUI-25 | **0.21±0.02** | **610±18** | **1.22±0.08** | 0.14±0.01 | **0.82±0.03** | **0.78±0.04** | 4.5 | 5.2 |
| NeuV-SLAM + IG | DUI-25 | 0.35±0.04 | 820±40 | 0.95±0.09 | **0.12±0.02** | 0.35±0.05 | 0.25±0.06 | 5.1 | − |
| SEAL | DUI-25 | 0.65±0.08 | 950±55 | 0.68±0.10 | 0.45±0.06 | 0.71±0.04 | 0.18±0.05 | 3.8 | − |
| LIMO + NBV | DUI-25 | 0.78±0.07 | 1150±60 | 0.55±0.11 | 0.21±0.04 | 0.20±0.06 | 0.15±0.07 | 3.5 | − |
| EGE | DUI-25 | 0.92±0.09 | 1380±70 | 0.41±0.13 | 0.65±0.08 | 0.15±0.04 | 0.41±0.08 | 1.8 | 2.1 |
| NBVP-FKIE | DUI-25 | 1.15±0.12 | >1800 | 0.25±0.15 | 0.82±0.11 | 0.10±0.05 | 0.32±0.09 | **1.1** | 1.5 |
| *Ablation Study in Multi-Agent Rescue Mission (MARM-25)* | | | | | | | | | |
| **SCENE (Ours)** | MARM-25 | **0.28±0.03** | **450±20** | **0.92±0.07** | 0.23±0.02 | **0.89±0.02** | − | 4.5 | 5.5 |
| SCENE-Decentralized | MARM-25 | 0.48±0.06 | 790±28 | 0.55±0.09 | 0.40±0.04 | 0.70±0.04 | − | 4.1 | **0.0** |
| SCENE-NoSem | MARM-25 | 0.61±0.07 | 1420±45 | 0.39±0.10 | 0.28±0.03 | 0.42±0.06 | − | 4.4 | 5.4 |
| SCENE-NoAoI | MARM-25 | 0.34±0.04 | 510±22 | 0.86±0.08 | 0.25±0.02 | 0.88±0.03 | − | 4.5 | 5.5 |
| SCENE-Geo | MARM-25 | 0.69±0.08 | 1650±60 | 0.33±0.12 | 0.30±0.04 | 0.28±0.07 | − | 4.2 | 5.3 |

- *Task Completion Time*: Time in seconds to complete the mission objective (e.g., finding all targets or exploring 95% of the volume).
- **Auxiliary Metrics**:
  - *Geometric Accuracy*: Root Mean Square Error (RMSE) of the reconstructed map.
  - *Semantic Accuracy*: Mean Intersection over Union (mIoU) for semantic segmentation.
  - *Dynamic Tracking Accuracy*: Multi-Object Tracking Accuracy (MOTA).
  - *Computational Load*: Giga FLoating-point Operations Per Second (GFLOPS) per agent.
  - *Coordination Overhead*: Percentage of computation time spent on GNN message passing.

## 4.4 RESULTS AND ANALYSIS

### 4.4.1 COMPREHENSIVE PERFORMANCE AND ABLATION STUDY

We first conducted a comprehensive evaluation in the dynamic *DUI-25* and task-driven *MARM-25* scenarios. Aggregated results are in Table 1. The results unequivocally demonstrate SCENE's superiority, as it achieves the lowest Cognitive Dissonance (CD), fastest Task Time, and highest Energy Efficiency (EE). This confirms our unified free-energy principle guides the team to not only a more accurate final map but also the most efficient path to achieve it.

Classic methods like *EGE* and *NBVP-FKIE* performed poorly, highlighting the limitations of purely geometric exploration. Modern SOTAs like *LIMO* and *SEAL*, while strong in their specialties, lacked a unified framework to achieve top overall performance. The ablation studies in MARM-25 are particularly insightful: disabling semantic guidance (*SCENE-NoSem*) or coordination (*SCENE-Decentralized*) drastically increased task time by over 215% and 75% respectively. This validates our core hypothesis that unifying objectives within a collaborative framework is essential. A detailed analysis, including visualizations of the cognitive free energy's evolution and the system's dynamic perception capabilities, is provided in Appendix C to further illustrate these findings.

### 4.4.2 EXPLAINABILITY AND EMERGENT COLLABORATION

Beyond performance metrics, it is crucial to understand *how* SCENE's internal logic translates to intelligent behavior. Figure 1 provides an interpretable view into the decision-making process. While geometric gain (a) provides a general drive to explore, the discovery of a potential semantic anomaly generates a strong, localized semantic gain signal (b). The final utility map (c) visualizes the principled trade-off: the resulting decision is intelligently directed towards a location that optimally reduces the *unified* information gain. This demonstrates how SCENE arbitrates between competing objectives, a foundational mechanism for its intelligent behavior.

This decision-making extends to the multi-agent level, as visualized in Figure 2. The dynamic GNN attention weights reveal the emergence of a sophisticated collaborative strategy. Initially (a), UGVs attend to UAVs, leveraging their aerial perspective. Upon an anomaly sighting by a UAV (b), the team's attentional focus rapidly shifts. This culminates in a dynamic "hand-off" (c), where attention

### Explainability of SCENE's Unified Decision-Making @ MARM-25

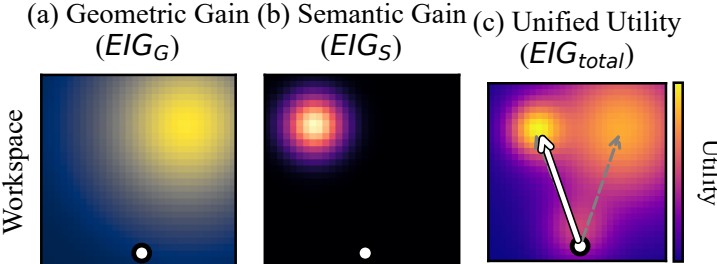

Figure 1: **Explainability of SCENE's Unified Decision-Making @ MARM-25.** Each panel displays a spatial utility heatmap, where brighter colors indicate higher expected information gain (EIG). The white circle marks the robot's current position. **(a)** A broad region of geometric uncertainty (*Geometric Gain*) encourages general exploration. **(b)** A concentrated peak of semantic novelty (*Semantic Gain*) signals a specific object of interest. **(c)** The resulting *Unified Utility* map demonstrates the principled trade-off. The final action (solid white arrow) is an intelligent compromise, distinct from greedy actions that would solely target maximum geometric gain (dashed gray arrow) or semantic gain (dashed gray arrow). This visualizes how SCENE arbitrates between competing objectives to achieve a globally optimal decision.

Figure 2: **Emergence of Heterogeneous Collaborative Strategy @ MARM-25.** This sequence visualizes the dynamic Graph Neural Network (GNN) attention weights between robots during a mission, where nodes represent robots (circles: UGVs, squares: UAVs) and edge thickness/color intensity corresponds to the attention strength between them. **(a)** During the initial *UAV-led broad search*, ground vehicles (UGVs) primarily attend to the scouting aerial vehicles (UAVs), leveraging their superior viewpoint. **(b)** Upon *anomaly sighting by a UAV* (UAV-7), the team's attentional focus rapidly converges onto the discovering agent. **(c)** A sophisticated *UGV re-tasked for confirmation* strategy emerges, where attention becomes highly concentrated between the discovering UAV and the closest UGV (UGV-3), forming a dynamic "hand-off" for ground-level inspection. This demonstrates how SCENE learns synergistic, role-differentiated behaviors rather than relying on predefined rules.

becomes highly focused between the discovering UAV and the closest UGV re-tasked for confirmation. This visualization explains the performance gap between the full SCENE framework and the *SCENE-Decentralized* ablation in Table 1; our method learns truly synergistic, role-differentiated strategies.

Table 2: Unified Robustness and Scalability Evaluation Under Extreme Conditions. A '–' indicates a test type without a specific tunable parameter.

| Method | Test Type | Parameter | ATE (cm) ↓ | Task Time (s) ↓ | Success Rate (%) ↑ |
|---|---|---|---|---|---|
| *Test 1: Perceptual Degradation in Subterranean Environment (BCE-25)* | | | | | |
| **SCENE (Ours)** | Perceptual Degradation | – | **1.1±0.3** | **950±40** | **97** |
| NeuV-SLAM + IG | Perceptual Degradation | – | 2.5±0.5 | 1180±55 | 94 |
| LIMO + NBV | Perceptual Degradation | – | 4.2±0.8 | 1520±70 | 85 |
| EGE | Perceptual Degradation | – | >10.0 | >2000 | 60 |
| *Test 2: Network Failure in Multi-Agent Rescue Mission (MARM-25)* | | | | | |
| **SCENE (Ours)** | Communication Interruption | 50% Loss | **2.5±0.4** | **580±28** | **95** |
| NeuV-SLAM + IG | Communication Interruption | 50% Loss | 4.8±0.7 | 1150±60 | 85 |
| EGE | Communication Interruption | 50% Loss | >10.0 | 1450±80 | 70 |
| **SCENE (Ours)** | Robot Dropout | 25% Team Loss (under 50% Comm. Loss) | **3.2±0.5** | **690±35** | **92** |
| NeuV-SLAM + IG | Robot Dropout | 25% Team Loss (under 50% Comm. Loss) | 6.5±1.0 | >1800 | 75 |
| *Test 3: Scalability in Multi-Agent Rescue Mission (MARM-25)* | | | | | |
| **SCENE (Ours)** | Scalability | 16 Robots | – | **155±12** | **98** |
| NeuV-SLAM + IG | Scalability | 16 Robots | – | 380±25 | 94 |
| EGE | Scalability | 16 Robots | – | 280±18 | 95 |

Table 3: Quantitative Analysis of Heterogeneous Strategy Emergence in MARM-25

| Metric | Homo. UGV (8 Robots) | Homo. UAV (8 Robots) | Hetero. (6U+2A) (Ours) |
|---|---|---|---|
| ***Primary Task Performance*** | | | |
| Task Time (s) ↓ | 1050±50 | 1220±65 | **450±20** |
| Tgt. Disc. Latency (s) ↓ | 580±35 | **330±28** | 350±30 |
| Conf. Accuracy (%) ↑ | 85±5.0 | 68±6.5 | **96±2.5** |
| ***Behavioral & Efficiency Metrics*** | | | |
| Total Distance (km) ↓ | 13.1±0.9 | 26.5±1.4 | **10.2±0.7** |
| Redund. View. (%) ↓ | 19.2±1.6 | 26.1±2.2 | **9.8±1.2** |
| Inter-Class Attn. | N/A | N/A | **0.67** |

### 4.4.3 ROBUSTNESS, SCALABILITY, AND HETEROGENEITY

A practical system must be resilient. We tested all methods under extreme conditions, with results consolidated in Table 2. SCENE demonstrates exceptional robustness, achieving the lowest Absolute Trajectory Error (ATE) in the perceptually-degraded BCE-25 cave. Network failure tests further underscore its resilience; its performance degrades gracefully, in stark contrast to methods like EGE. Further analysis, including a performance degradation surface and scalability plot, is available in Appendix C.

Finally, we quantified the benefits of heterogeneity. As shown in Table 3, our default mixed team (6 UGVs, 2 UAVs) drastically outperforms homogeneous teams of 8 UGVs or 8 UAVs in the 3D MARM-25 scenario. The heterogeneous team achieves the best of both worlds: fast discovery from UAVs and high-accuracy confirmation from UGVs, leading to the fastest task completion and highest efficiency. The high "Inter-Class Attention" metric (0.67) confirms strong information flow between UAVs and UGVs, proving that SCENE orchestrates a true heterogeneous collective intelligence.

## 5 CONCLUSION

We introduced SCENE, a novel framework that unifies multi-robot exploration, semantic search, and dynamic tracking under the principle of cognitive free-energy minimization. Our extensive experiments demonstrate that this unified approach facilitates emergent collaboration and achieves superior efficiency and robustness against strong baselines in complex missions. Future work includes deploying SCENE on physical robots for real-world validation and extending its cognitive model to incorporate higher-level causal reasoning and planning.

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

## A  THEORETICAL AND METHODOLOGICAL DETAILS

### A.1  DETAILED DERIVATION OF EIG COMPONENTS

Let the team's belief over the world state $M$ at time $t$ be represented by the posterior distribution $p(M|Z_t)$, where $Z_t = \{z_1, ..., z_t\}$ is the history of observations. The objective is to select a collective action $a$ that maximizes the EIG, defined as the mutual information between the future observation $z_{t+1}$ and the world state $M$:

$$a^* = \arg\max_a I(M; z_{t+1}|a, Z_t) \tag{13}$$

This can be expressed as the expected reduction in the entropy of the world model upon taking a new measurement:

$$a^* = \arg\max_a \left( H(M|Z_t) - \mathbb{E}_{z_{t+1}|a,Z_t}[H(M|Z_t, z_{t+1})] \right) \tag{14}$$

The pivotal innovation of our work lies in the definition of the world state $M$. We propose that $M$ is not merely a geometric map, but a unified cognitive state encompassing **Geometry (G)**, **Semantics (S)**, and **Dynamic Timeliness (D)**.

Substituting this composite state $M = \{G, S, D\}$ into the mutual information term allows us to apply the chain rule of information theory for an exact, but computationally challenging, expansion:

$$I(M; z_{t+1}) = I(\{G, S, D\}; z_{t+1}) = I(G; z_{t+1}) + I(S; z_{t+1}|G) + I(D; z_{t+1}|G, S) \tag{15}$$

where all terms are implicitly conditioned on the action $a$ and history $Z_t$. While theoretically sound, this formulation is intractable for real-time decision-making, as it requires evaluating complex conditional information gains (e.g., the information gained about semantics, given that the geometry is already fully known from the same observation).

**The Principled Approximation via Conditional Independence.** To derive a tractable objective, we introduce a key approximation: the conditional independence of the information gains. We posit that, given the agent's current belief state, the new information an observation $z_{t+1}$ provides about one component (e.g., Semantics) is approximately independent of the information it provides about another (e.g., Geometry). This implies:

$$I(S; z_{t+1}|G) \approx I(S; z_{t+1}) \tag{16}$$

$$I(D; z_{t+1}|G, S) \approx I(D; z_{t+1}) \tag{17}$$

The rationale is that while the world properties themselves are coupled (e.g., an object's shape influences its semantic label), the \*process of reducing uncertainty\* about them with a single new sensor reading can be treated as largely parallel. For instance, an RGB-D frame simultaneously provides evidence for both surface locations (from depth) and surface appearance (from color), allowing for a decoupled estimation of information gain. This approximation simplifies the chain rule in Eq. equation 15 into a practical, additive form:

$$\text{EIG}(a) \approx I(G; z_{t+1}) + I(S; z_{t+1}) + I(D; z_{t+1}) \tag{18}$$

This leads directly to the formulation where the total EIG is a sum of the individual expected information gains for each component.

Assuming conditional independence between these components given the observations, the total information gain can be additively decomposed. We formalize our unified objective as minimizing the negative EIG, which constitutes our cognitive free energy $\mathcal{F}$:

$$\mathcal{F}(a) = -\text{EIG}(a) \tag{19}$$

Each component in Equation 1 from the main text represents a distinct cognitive drive:

1. **Geometric Information Gain ($\text{EIG}_G$):** This term measures the expected reduction in *geometric uncertainty*. It quantifies the value of an action in refining the agent's knowledge of the physical structure of the environment—resolving questions about where surfaces exist versus where free space lies. This drives the classic impulse to map unknown areas.
2. **Semantic Information Gain ($\text{EIG}_S$):** This term measures the expected *semantic surprise* or novelty. A high gain results from observations that are highly improbable under the current semantic model, such as encountering an object category with high model uncertainty or one that contradicts the environmental prior. This drives the robot to discover and scrutinize meaningful, novel entities.
3. **Dynamic Information Gain ($\text{EIG}_D$):** This term measures the expected contribution to maintaining *knowledge freshness*. It quantifies the value of reducing uncertainty caused by the passage of time in a dynamic world. By revisiting areas where information is likely to be stale (high Age of Information), the agent actively counteracts cognitive entropy, which is crucial for maintaining situational awareness in non-stationary environments.

## A.2 COGNITIVE FREE ENERGY: FROM THEORY TO A COMPUTABLE OBJECTIVE

To operationalize the principle of cognitive free-energy minimization, we must define how the abstract information density terms—geometric, semantic, and dynamic—are instantiated as concrete,

computable functions of our neural world representation. This section details these instantiations and describes the numerical procedure for evaluating the total Expected Information Gain (EIG) of any candidate action.

### A.2.1 Instantiation of Information Density Functions

Each component of the information gain is derived directly from the outputs and internal states of the neural fields $\Phi_\theta$ and $\Psi_\xi$.

**Geometric Information Density ($\mathcal{I}_G$):** The geometric uncertainty at a point $\mathbf{x}$ is quantified by the epistemic uncertainty in the model's TSDF prediction. We approximate this as the running variance of TSDF predictions for that location over time. The multi-resolution hash grid of $\Phi_\theta$ provides a natural structure to store these statistics. For each feature vector stored at each level of the hash grid, we associate a set of running statistics $(n, \mu, M_2)$, representing the observation count, mean, and the sum of squares of differences from the mean, respectively. These are updated efficiently for each new TSDF prediction using Welford's online algorithm. The final variance at a point $\mathbf{x}$, $\sigma_g^2(\mathbf{x}, t)$, is obtained via the same trilinear interpolation process used for feature retrieval. Thus, the geometric information density is formally defined as:

$$\mathcal{I}_G(\mathbf{x}, t) = \sigma_g^2(\mathbf{x}, t) \tag{20}$$

This formulation directly drives agents to regions where the model's own structural knowledge is inconsistent or lacking.

**Semantic Information Density ($\mathcal{I}_S$):** The total semantic information density is a weighted combination of epistemic uncertainty regarding known classes and novelty corresponding to unknown entities.

$$\mathcal{I}_S(\mathbf{x}, t) = \lambda_e \mathcal{I}_S^{\text{epistemic}}(\mathbf{x}, t) + \lambda_n \mathcal{I}_S^{\text{novelty}}(\mathbf{x}, t) \tag{21}$$

The two components are calculated as follows:

- **Epistemic Uncertainty ($\mathcal{I}_S^{\text{epistemic}}$):** This term, calculated from the deep ensemble's outputs as defined in Eq. equation 5 of the main paper, captures the model's disagreement about the identity of an object within its known semantic categories. A high value signifies a location that is confusing to the model.
- **Open-World Novelty ($\mathcal{I}_S^{\text{novelty}}$):** This term, defined in Eq. equation 6, measures the out-of-distribution score for a new observation. The feature vector $\mathbf{f}_s(\mathbf{x}, t)$ used for this calculation is the penultimate feature vector from the semantic MLP heads, averaged across the ensemble for stability. A high value indicates that the observed features are statistically dissimilar from all known class prototypes, signaling a potentially novel object.

The hyperparameters are set to $\lambda_e = 1.0$ and $\lambda_n = 0.5$, prioritizing the reduction of uncertainty within the existing world model while maintaining sensitivity to genuinely new semantic information.

**Dynamic Information Density ($\mathcal{I}_D$):** The dynamic information density directly quantifies the value of reducing time-induced uncertainty. This is elegantly and simply represented by the current predicted Age of Information from the AoI field $\Psi_\xi$.

$$\mathcal{I}_D(\mathbf{x}, t) = A_\xi(\mathbf{x}, t) \tag{22}$$

This formulation is principled: the potential information gain from revisiting a location is precisely the amount of accumulated uncertainty (staleness) that the new observation will collapse.

### A.2.2 Numerical Approximation of the Action-Contingent EIG

The final objective is to select an action $a$ that maximizes the total EIG, as formulated in Eq. equation 2. An action $a$ is defined by a candidate pose $\mathbf{T}_{\text{cand}} \in SE(3)$ to be reached by the robot. The integral over the prospective field of view $\mathcal{V}_a$ is computed numerically via Monte Carlo integration. The procedure is as follows:

1. **Point Cloud Sampling:** We sample a set of $N_s = 2048$ points, $X_a = \{\mathbf{x}_j\}_{j=1}^{N_s}$, within the camera frustum defined by the candidate action's pose $\mathbf{T}_{\text{cand}}$.

2. **Visibility Estimation:** For each point $\mathbf{x}_j \in X_a$, we estimate its visibility $V(\mathbf{x}_j|a) \in \{0, 1\}$. This is done by performing ray-marching from the camera origin of $\mathbf{T}_{\text{cand}}$ towards $\mathbf{x}_j$ through the current TSDF field predicted by $\Phi_\theta$. If the ray intersects a surface (a zero-crossing of the TSDF) before reaching a threshold distance from $\mathbf{x}_j$, the point is considered occluded, and $V(\mathbf{x}_j|a) = 0$. Otherwise, $V(\mathbf{x}_j|a) = 1$.

3. **EIG Computation:** The total EIG for action $a$ is approximated as the weighted sum of information densities over all visible sampled points:

$$\text{EIG}(a) \approx \frac{1}{\sum V(\mathbf{x}_j|a)} \sum_{j=1}^{N_s} V(\mathbf{x}_j|a) \left(w_g \mathcal{I}_G(\mathbf{x}_j, t) + w_s \mathcal{I}_S(\mathbf{x}_j, t) + w_d \mathcal{I}_D(\mathbf{x}_j, t)\right) \quad (23)$$

The task-dependent weights $w_g, w_s, w_d$ modulate the exploration policy's priorities according to the mission context, as detailed in Table 4. This computable function provides the core signal for both individual action proposal in the decision-making pipeline and the global reward for MARL training.

Table 4: Task-dependent weights for EIG computation, reflecting mission priorities.

| Scenario | $w_g$ | $w_s$ | $w_d$ | Rationale |
|---|---|---|---|---|
| DUI-25 | 0.3 | 0.3 | 0.4 | Balanced, with a slight emphasis on dynamic timeliness. |
| MARM-25 | 0.4 | 0.6 | 0.0 | Emphasis on semantic discovery; no dynamic elements. |
| BCE-25 | 0.7 | 0.3 | 0.0 | Strong emphasis on geometric mapping in sparse environments. |

# B  IMPLEMENTATION AND ARCHITECTURAL DETAILS

## B.1  DETAILED SCENE PIPELINE

The SCENE framework's perception-planning-action loop operates in continuous cycles as follows:

**Step 1: Building the Unified World Representation.** The team collaboratively constructs a shared world model using a **hash-encoded multi-head neural implicit field**. This continuous function, $\Phi_\theta(x, t)$, is queried at any spatio-temporal coordinate and has three specialized output heads:

- **Geometric Field:** Provides TSDF values for high-fidelity surface geometry.
- **Semantic Field:** Outputs a probability distribution over object classes, including associated uncertainty.
- **Dynamic Field:** Predicts the near-future probability of state changes in the environment.

This unified field is trained online and self-supervisedly by all robots, ensuring a continuously refined and compact world representation $\theta$.

**Step 2: Quantifying the Cognitive State.** We quantify the team's understanding of the world via a scalar metric: the **cognitive free energy, $\mathcal{F}$.** At each timestep, its three key components are computed from the world model $\theta$:

- **Global Geometric Uncertainty $\mathcal{U}_{total}$:** Derived from the global variance of the TSDF predictions.
- **Global Semantic Novelty $\mathcal{S}_{total}$:** Measured using a **prototype-based open-world semantic memory**, which detects novel or unexpected objects.
- **Global Information Age $\mathcal{D}_{total}$:** Calculated as the global integral of a dedicated **Age of Information (AoI) neural field** Shi et al. (2020) that tracks data freshness.

**Step 3: Collaborative Action Generation and Evaluation.** Each robot's decision-making involves two stages:

1. *Intent Generation:* Each robot $i$ samples candidate actions and simulates their expected individual information gain. The top $K$ actions are encoded as **action intents**, $\mathcal{I}_i$, for broadcasting.
2. *Collaborative Evaluation:* Robots exchange intents with neighbors. A **Graph Neural Network (GNN)** with a learnable attention mechanism processes this information, allowing each robot to reason about team-level synergy. The GNN refines the utility of each action

to maximize the reduction in the *team's* total free energy, producing a final collaborative Q-value.

**Step 4: Execution and Learning.** The robot executes the action with the highest Q-value. The entire system is trained end-to-end within a **Multi-Agent Reinforcement Learning (MARL)** framework. The team's **shared reward signal** is the **actual measured decrease in global free energy** $\mathcal{F}$. This directly incentivizes the policy to learn efficient, non-redundant, and collaborative exploration behaviors. This perception-planning-action loop drives the team to relentlessly convert environmental unknowns into known, fresh, and semantically meaningful cognition.

## B.2 Unified World Representation: Data Flow and Self-Supervised Learning

The unified world representation, consisting of the primary field $\Phi_\theta$ and the Age of Information (AoI) field $\Psi_\xi$, is trained online and continuously from the distributed sensor streams of the multi-robot team. Here, we detail the end-to-end data flow and the precise formulation of the loss functions that enable self-supervised learning.

### B.2.1 End-to-End Data Flow

For each robot $i$ at every timestep $t$, the perception and modeling pipeline is executed as follows:

1. **Data Acquisition:** The robot captures an RGB-D image, $I_t \in \mathbb{R}^{H \times W \times 4}$, and its corresponding pose, $\mathbf{T}_t \in SE(3)$, which is estimated and continuously refined by a lightweight, onboard Visual-Inertial Odometry (VIO) frontend.
2. **Ray and Point Sampling:** A batch of $N_r$ rays (typically $N_r = 1024$) is randomly sampled from the pixels of the current image. Along each ray $\mathbf{r}$, we sample $N_p$ points (typically $N_p = 64$) through a coarse-to-fine hierarchical sampling strategy. This strategy stratifies the sampling along the ray and then performs importance sampling based on a coarse model's volume density to concentrate points near expected surfaces.
3. **Training Datapoint Generation:** Each sampled 3D point $\mathbf{x}$ in the world frame, along with the current timestamp $t$, forms a spatio-temporal coordinate $(\mathbf{x}, t)$. This coordinate, paired with its corresponding ground-truth color $\mathbf{c}_{gt}(\mathbf{x})$, depth $d_{gt}(\mathbf{x})$, and projected 2D semantic pseudo-label $l_{gt}(\mathbf{x})$, constitutes a single training datapoint. These datapoints are used to jointly optimize the parameters $\theta$ of $\Phi_\theta$ and $\xi$ of $\Psi_\xi$.

### B.2.2 Self-Supervised Loss Functions

The entire representation is trained via a composite loss function $\mathcal{L}_{\text{total}}$, which is a weighted sum of four specialized, self-supervised loss components.

**Geometric Loss ($\mathcal{L}_{\text{geom}}$):**  The geometric head of $\Phi_\theta$ is trained to predict the Truncated Signed Distance Function (TSDF) value for any given spatio-temporal coordinate. The supervision is directly derived from the aligned depth map. For a set of sampled points $X$ along the rays, the geometric loss is a weighted L1 loss between the predicted TSDF value and the ground-truth TSDF value.

$$\mathcal{L}_{\text{geom}} = \frac{1}{|X|} \sum_{\mathbf{x} \in X} w(\mathbf{x}) \cdot |d_\theta(\mathbf{x}, t) - d_{gt}(\mathbf{x})| \tag{24}$$

where $d_\theta(\mathbf{x}, t)$ is the TSDF value output by the geometric head. The ground-truth value $d_{gt}(\mathbf{x})$ is computed as $d_{gt}(\mathbf{x}) = \text{clip}(\|\mathbf{p}_c(\mathbf{x})\|_2 - D_t(\pi(\mathbf{p}_c(\mathbf{x}))), -\tau_{\text{trunc}}, \tau_{\text{trunc}})$, where $\mathbf{p}_c(\mathbf{x})$ is the point $\mathbf{x}$ in the camera frame, $\pi(\cdot)$ is the projection function, $D_t$ is the depth map, and $\tau_{\text{trunc}}$ is the truncation distance (set to 5cm). The weight $w(\mathbf{x}) = \exp(-\alpha|d_{gt}(\mathbf{x})|)$ with $\alpha = 10.0$ prioritizes points near the zero-level crossing, ensuring high fidelity surface reconstruction.

**Semantic Loss ($\mathcal{L}_{\text{sem}}$):**  The $N_e$ independent heads of the semantic deep ensemble are supervised using 2D pseudo-labels projected into 3D. A pre-trained, fixed 2D foundation model (e.g., SAM) provides a semantic label for each pixel in the input image. To ensure temporal consistency and handle projection ambiguities, we maintain a frequency-based histogram of labels for each voxel in a coarse grid. The supervision label $l_\mathbf{x}$ for a point $\mathbf{x}$ is the mode of the histogram in its corresponding

voxel. The semantic loss is the sum of standard cross-entropy losses over all ensemble heads:

$$\mathcal{L}_{\text{sem}} = \sum_{j=1}^{N_e} \frac{1}{|X|} \sum_{\mathbf{x} \in X} \text{CrossEntropy}(p_j(\mathbf{x}, t), l_{\mathbf{x}}) \tag{25}$$

where $p_j(\mathbf{x}, t)$ is the softmax probability vector from the $j$-th semantic head. This trains all heads on the same temporally-stabilized target, with their diverse predictions arising from different random initializations.

**Dynamic Loss ($\mathcal{L}_{\text{dyn}}$):** The dynamic head is trained to predict the probability of future state changes, supervised by detecting temporal inconsistencies in the learned representation itself. For a point $\mathbf{x}$ observed at time $t_1$ and revisited at time $t_2$, we generate a positive change label $c_{\mathbf{x}} = 1$ if the L2 distance between its geometric feature vectors $\mathbf{f}_g$ from the geometric head exceeds a threshold $\tau_{\text{dyn}}$ (i.e., $\|\mathbf{f}_g(\mathbf{x}, t_2) - \mathbf{f}_g(\mathbf{x}, t_1)\|_2 > \tau_{\text{dyn}}$). Otherwise, $c_{\mathbf{x}} = 0$. The loss is a binary cross-entropy applied to the earlier prediction:

$$\mathcal{L}_{\text{dyn}} = -\frac{1}{|X_{\text{revisit}}|} \sum_{\mathbf{x} \in X_{\text{revisit}}} [c_{\mathbf{x}} \log(v_\theta(\mathbf{x}, t_1)) + (1 - c_{\mathbf{x}}) \log(1 - v_\theta(\mathbf{x}, t_1))] \tag{26}$$

where $v_\theta(\mathbf{x}, t_1)$ is the change probability predicted by the dynamic head at the first observation time $t_1$. This incentivizes the model to learn to anticipate changes based on past observations.

**Age of Information Loss ($\mathcal{L}_{\text{AoI}}$):** The AoI field $\Psi_\xi$ learns the continuous AoI function $A(\mathbf{x}, t)$ through a unique, physics-informed supervisory signal. The loss function enforces two fundamental properties of AoI: (1) information is perfectly fresh upon observation, and (2) information ages linearly with time elsewhere. This is formulated as a two-part MSE loss:

$$\mathcal{L}_{\text{AoI}} = \frac{1}{|X_{\text{view}}|} \sum_{\mathbf{x} \in X_{\text{view}}} |A_\xi(\mathbf{x}, t) - 0|^2 + \frac{1}{|X_{\text{mem}}|} \sum_{\mathbf{x}' \in X_{\text{mem}}} |A_\xi(\mathbf{x}', t) - (\text{sg}[A_\xi(\mathbf{x}', t - \Delta t)] + \Delta t)|^2 \tag{27}$$

where $X_{\text{view}}$ are points sampled from the current sensor frustum, and $X_{\text{mem}}$ are points sampled randomly from the entire represented scene volume. The 'sg[·]' operator denotes a stop-gradient, ensuring that the second term's target, derived from the previous state of the network, is treated as a fixed constant for stable training.

### B.2.3 TOTAL TRAINING LOSS

The parameters $\theta$ and $\xi$ of the neural fields are jointly optimized by minimizing a weighted sum of the aforementioned loss components.

$$\mathcal{L}_{\text{total}} = w_g \mathcal{L}_{\text{geom}} + w_s \mathcal{L}_{\text{sem}} + w_d \mathcal{L}_{\text{dyn}} + w_a \mathcal{L}_{\text{AoI}} \tag{28}$$

The loss weights are fixed hyperparameters, set to balance the learning dynamics and relative importance of each cognitive component. Based on empirical validation, we use the values provided in Table 5.

Table 5: Loss function weights for the joint optimization of the neural world representation.

| Weight | Value | Associated Component |
|--------|-------|----------------------|
| $w_g$ | 1.0 | Geometric Loss ($\mathcal{L}_{\text{geom}}$) |
| $w_s$ | 0.5 | Semantic Loss ($\mathcal{L}_{\text{sem}}$) |
| $w_d$ | 1.5 | Dynamic Loss ($\mathcal{L}_{\text{dyn}}$) |
| $w_a$ | 0.1 | Age of Information Loss ($\mathcal{L}_{\text{AoI}}$) |

### B.3 OPEN-WORLD SEMANTIC MODULE: PROTOTYPICAL MEMORY AND NOVELTY DETECTION

The open-world semantic capability of SCENE is enabled by a Prototypical Network-based memory module that operates in conjunction with the main semantic field. While the deep ensemble of the semantic head in $\Phi_\theta$ provides uncertainty estimates over a fixed set of $K$ known classes, this module allows each agent to dynamically track class feature distributions and detect entirely novel object categories. Each agent maintains its own instance of this memory, which is synchronized with teammates periodically.

The core of the module is a set of class prototypes, $\{\mathcal{P}_k\}_{k=1}^{K}$, where each prototype $\mathcal{P}_k = \{\mathbf{p}_k, \mathbf{\Sigma}_k, n_k\}$ consists of the mean feature vector $\mathbf{p}_k \in \mathbb{R}^D$, the covariance matrix $\mathbf{\Sigma}_k \in \mathbb{R}^{D \times D}$, and the observation count $n_k$ for class $k$. The feature vectors are extracted from the penultimate layer of the semantic MLP heads, averaged across the ensemble for stability.

### B.3.1 PROTOTYPE INITIALIZATION

A prototype for a known class $k$ is not pre-defined but is initialized online once sufficient evidence has been gathered. This "late initialization" strategy ensures that the prototypes are representative of in-situ observations rather than being biased by pre-training data.

The initialization for class $k$ is triggered when a minimum number of high-confidence observations, $M_{\text{init}}$, have been collected for that class. We set $M_{\text{init}} = 20$. Specifically, for each new observation with a semantic feature vector $\mathbf{f}_s$ and a predicted class label $k$ (with softmax confidence $> \tau_{\text{conf}}$), we add $\mathbf{f}_s$ to a temporary buffer $\mathcal{B}_k$. Once $|\mathcal{B}_k| \geq M_{\text{init}}$, the prototype $\mathcal{P}_k$ is initialized by computing the sample mean and covariance from the buffered features:

$$\mathbf{p}_k = \frac{1}{M_{\text{init}}} \sum_{\mathbf{f}_s \in \mathcal{B}_k} \mathbf{f}_s \tag{29}$$

$$\mathbf{\Sigma}_k = \frac{1}{M_{\text{init}} - 1} \sum_{\mathbf{f}_s \in \mathcal{B}_k} (\mathbf{f}_s - \mathbf{p}_k)(\mathbf{f}_s - \mathbf{p}_k)^T + \epsilon I \tag{30}$$

where $\epsilon I$ is a small regularization term (with $\epsilon = 10^{-6}$) added to the diagonal to ensure the covariance matrix is positive definite and invertible. After initialization, the buffer $\mathcal{B}_k$ is cleared.

### B.3.2 ONLINE COVARIANCE-PRESERVING UPDATE

Once initialized, the prototypes are updated online with each new observation to adapt to variations in object appearance and environmental conditions. A simple moving average can be numerically unstable and does not correctly update the covariance. Instead, we employ a numerically stable, rank-one update method derived from Welford's online algorithm for computing variance, extended to the multivariate case for covariance.

Given a prototype $\{\mathbf{p}_k, \mathbf{\Sigma}_k, n_k\}$ and a new feature vector observation $\mathbf{f}_{\text{new}}$ for class $k$, the update proceeds as follows:

1. Increment the count: $n_{k,\text{new}} = n_k + 1$.
2. Update the mean: $\mathbf{p}_{k,\text{new}} = \mathbf{p}_k + \frac{1}{n_{k,\text{new}}}(\mathbf{f}_{\text{new}} - \mathbf{p}_k)$.
3. Update the covariance matrix using the previous and new means:

$$\mathbf{\Sigma}_{k,\text{new}} = \frac{n_k - 1}{n_{k,\text{new}} - 1}\mathbf{\Sigma}_k + \frac{n_k}{n_{k,\text{new}}(n_{k,\text{new}} - 1)}(\mathbf{f}_{\text{new}} - \mathbf{p}_k)(\mathbf{f}_{\text{new}} - \mathbf{p}_k)^T \tag{31}$$

This method provides a robust and efficient way to maintain accurate second-order statistics of the class feature distributions without storing all past observations.

### B.3.3 NEW CATEGORY DETECTION AND PROMOTION

The detection of novel classes is based on the Mahalanobis distance, which serves as our novelty score $\mathcal{S}_{\text{novelty}}(\mathbf{f}_s)$. For a new feature vector $\mathbf{f}_s$, this score is its distance to the closest known class prototype:

$$\mathcal{S}_{\text{novelty}}(\mathbf{f}_s) = \min_{k=1,\ldots,K} \sqrt{(\mathbf{f}_s - \mathbf{p}_k)^T \mathbf{\Sigma}_k^{-1} (\mathbf{f}_s - \mathbf{p}_k)} \tag{32}$$

A high score indicates that the observation is an outlier with respect to all known class distributions. The detection process is designed as a two-stage mechanism to prevent spurious new categories from single noisy observations.

**Novelty Thresholding.** The novelty threshold, $\tau_{\text{novelty}}$, is not a fixed global value. Instead, it is class-dependent and derived from the statistics of the Chi-squared distribution. Assuming the features for class $k$ are approximately Gaussian, the squared Mahalanobis distance follows a Chi-squared distribution with $D$ degrees of freedom ($\chi_D^2$), where $D$ is the feature dimension. We set

the threshold for declaring a point an outlier relative to class $k$ as the critical value for a p-value of $p = 0.001$. The final novelty threshold is the minimum of these values over all initialized classes: $\tau_{\text{novelty}} = \min_k \sqrt{\chi_D^2(1-p)}$. This adaptive thresholding makes the novelty detection robust to classes with different feature variances.

**Candidate Promotion Mechanism.** When a spatial region consistently produces observations with novelty scores exceeding $\tau_{\text{novelty}}$, the system initiates a new class creation process:

1. **Candidate Generation:** If an observation $\mathbf{f}_s$ has $\mathcal{S}_{\text{novelty}}(\mathbf{f}_s) > \tau_{\text{novelty}}$, it is not immediately classified as a new object. Instead, it is stored in a global *candidate buffer*, $\mathcal{B}_{\text{cand}}$, and the corresponding map location is temporarily marked as a "region of interest."
2. **Spatial Clustering and Verification:** Periodically, we apply a density-based clustering algorithm (DBSCAN) to the feature vectors in $\mathcal{B}_{\text{cand}}$. If a stable cluster with at least $M_{\text{init}}$ points emerges, it is considered a high-confidence candidate for a new class.
3. **Prototype Promotion:** A candidate cluster that passes verification is promoted to a new, permanent class. A new prototype $\mathcal{P}_{K+1}$ is initialized from the cluster's members using the same procedure as described in Section B.3.1. The class counter is incremented ($K \leftarrow K + 1$), and the semantic head of the world model $\Phi_\theta$ can be optionally fine-tuned to include this new class if sufficient data is collected.

This robust, multi-stage process ensures that new classes are only created based on consistent and spatially coherent evidence, making the system's open-world perception both adaptive and reliable.

## B.4 ACTION SPACE AND CANDIDATE PROPOSAL GENERATION

The decision-making process in SCENE relies on evaluating a discrete set of candidate actions to select the one that maximizes the expected collaborative information gain. This section defines the action space and details the hybrid sampling strategy used to generate these candidates, ensuring a comprehensive yet computationally tractable exploration of potential future states.

### B.4.1 ACTION SPACE DEFINITION

The action space for each robot $i$ is defined as a discrete set of reachable target waypoints. An action $a_i$ corresponds to a target pose $\mathbf{T}_{\text{goal}} \in SE(3)$ to be reached within a fixed planning horizon. This formulation transforms the planning problem into a high-level decision-making task over a finite set of strategic options, which is well-suited for our Q-learning framework. The low-level path planning and control required to reach $\mathbf{T}_{\text{goal}}$ are delegated to a standard motion planner (e.g., RRT* followed by a trajectory optimizer), which also provides feasibility checks.

This choice of a discretized waypoint space, rather than a continuous velocity command space, offers two key advantages:

1. **Decoupling of Decision Horizons:** It allows the MARL policy to operate at a strategic level (where to go next) without being burdened by the complexities of micro-scale motion control.
2. **Tractability of EIG Evaluation:** It provides a concrete future viewpoint from which the integral for the Expected Information Gain (EIG) in Eq. equation 23 can be efficiently and accurately computed.

### B.4.2 HYBRID CANDIDATE SAMPLING STRATEGY

To ensure that the set of candidate actions is both diverse and relevant to the current cognitive state, we employ a hybrid sampling strategy that combines three distinct approaches. At each decision step, a total of $M = 64$ candidate waypoints are generated by drawing samples from these three pools.

**1. Information-Guided Sampling (Exploitation of Belief).** This strategy leverages the unified world model to actively sample waypoints in regions of high potential information gain. It focuses the search on areas the system already knows are "interesting."

- **Procedure:** We first generate a coarse voxel grid of the local environment around the robot. The value of each voxel is computed as the integrated information density (the integrand of Eq.

equation 2) within its volume. This creates a spatial probability distribution where high-value voxels represent regions of high uncertainty, novelty, or staleness.

- **Sampling:** We then sample $N_{ig} = 32$ waypoint candidates from this distribution. This biases the search towards promising regions, such as the far side of a partially mapped room (high geometric uncertainty) or near an object with ambiguous semantic labels (high semantic uncertainty). A reachability check using a straight-line path approximation is performed to discard clearly infeasible samples.

**2. Frontier-Based Sampling (Geometric Exploration).**  To ensure systematic exploration of unknown space, we incorporate a classic frontier-based approach. Frontiers are defined as the boundaries between known free space and unobserved space.

- **Procedure:** We perform a fast 3D segmentation on the current geometric map (derived from $\Phi_\theta$) to identify frontier voxels. These voxels are then clustered using DBSCAN.
- **Sampling:** We sample $N_f = 16$ waypoint candidates from the centroids of the largest frontier clusters. To generate valid viewpoints, the sampled point is moved back along its normal vector into free space, and the orientation is set to face the frontier. This method explicitly drives the geometric discovery aspect of the mission.

**3. RRT-Based Exploratory Sampling (Path-Aware Exploration).**  To ensure candidates are kinodynamically feasible and to encourage exploration into complex, non-convex spaces (e.g., corridors, tunnels), we use a path-aware sampling method based on Rapidly-exploring Random Trees (RRT).

- **Procedure:** We run multiple short, independent instances of RRT* from the robot's current state. The tree expansion is biased towards a combination of random directions and directions with high information gain. The search is terminated after a fixed number of iterations or a maximum path length is reached.
- **Sampling:** We select $N_{rrt} = 16$ candidates from the final nodes of the resulting RRT* branches. This ensures that every candidate is associated with a known-feasible path and encourages the discovery of non-obvious routes that would be missed by direct sampling methods. This is particularly crucial for UGV navigation in cluttered environments.

By combining these three strategies, SCENE generates a rich set of candidate actions at each step. This set effectively balances exploiting the current world model's belief state, systematically exploring unknown frontiers, and discovering kinodynamically feasible paths into novel regions, forming a robust foundation for the collaborative decision-making policy.

### B.5    Collaborative Decision-Making: Algorithmic Pipeline

The SCENE framework employs a Multi-Agent Reinforcement Learning (MARL) approach within the Centralized Training, Decentralized Execution (CTDE) paradigm. This allows the system to learn complex, synergistic behaviors in simulation while remaining computationally tractable for real-world deployment. This section details the decentralized execution algorithm run by each robot and subsequently describes the centralized training procedure.

#### B.5.1    Decentralized Execution Pipeline

At each decision step, every robot $i$ independently executes Algorithm 1 to select its next action. The algorithm is designed for efficiency, leveraging parallel computation for evaluating candidate actions and relying on a low-bandwidth exchange of "Action Intents" for coordination. The detailed pipeline is presented below.

#### B.5.2    Centralized Training Procedure

The parameters of the individual Q-networks ($Q_i$), the GAT, and the state-intent encoder (MLP$_{enc}$) are trained end-to-end in a centralized manner using the QMIX value decomposition architecture. The training process is as follows:

1. **Data Collection:** During simulated exploration, at each timestep $t$, the team executes actions based on the current policy. The tuple $(\mathbf{S}_t, \mathbf{A}_t, R_t, \mathbf{S}_{t+1})$ is stored in a shared replay

---

**Algorithm 1** SCENE - Detailed Collaborative Decision-Making Pipeline (Robot $i$)

---

1: **Input:** World model fields $\Phi_\theta, \Psi_\xi$; Local state embedding $\mathbf{s}_i \in \mathbb{R}^{77}$; GAT parameters $\Theta_{GAT}$; Q-network $Q_i$.

2: **Output:** Optimal collaborative action $a_i^* \in SE(3)$.

3:                 $\triangleright$ — *Phase 1: Individual Proposal Generation (Parallelized)* —

4: $\mathcal{A}_{\text{cand}} \leftarrow \text{GenerateCandidateActions}(M = 64, \text{robot\_pose}_i, \Phi_\theta)$    $\triangleright$ Hybrid sampling (see Appendix B.4)

5: $\mathcal{E}_{\text{ind}} \leftarrow \{\}$                   $\triangleright$ List of (action, EIG) pairs

6: **for** each action $a_j \in \mathcal{A}_{\text{cand}}$ **in parallel do**

7:      $\text{EIG}_j \leftarrow \text{ComputeEIG}(a_j, \Phi_\theta, \Psi_\xi)$      $\triangleright$ Numerical integration via Eq. equation 23

8:      Append $(a_j, \text{EIG}_j)$ to $\mathcal{E}_{\text{ind}}$.

9: **end for**

10:                 $\triangleright$ — *Phase 2: Intent Encoding and Communication* —

11: $\mathcal{E}_{\text{topK}} \leftarrow \text{TopK}(\mathcal{E}_{\text{ind}}, K = 5)$             $\triangleright$ Select top 5 based on EIG

12: $\mathcal{I}_i \leftarrow \text{EncodeIntent}(\mathcal{E}_{\text{topK}})$             $\triangleright$ Create 5x8 intent tensor

13: $\text{Broadcast}(\mathcal{I}_i)$; Receive $\{\mathcal{I}_j\}_{j \in \mathcal{N}_i}$ from neighbors $\mathcal{N}_i$.

14:                 $\triangleright$ — *Phase 3: Collaborative Context Generation with GAT* —

15: $V \leftarrow \{i\} \cup \mathcal{N}_i$             $\triangleright$ Nodes in the local communication graph

16: Initialize node features $\mathbf{H} \in \mathbb{R}^{|V| \times 128}$.

17: **for** each node $j \in V$ **in parallel do**

18:      $\mathbf{s}_j, \mathcal{I}_j \leftarrow \text{GetNodeStateAndIntent}(j)$

19:      $\mathbf{h}_j \leftarrow \text{MLP}_{\text{enc}}([\mathbf{s}_j || \text{Flatten}(\mathcal{I}_j)])$             $\triangleright$ Encode state and intent

20:      $\mathbf{H}[j] \leftarrow \mathbf{h}_j$

21: **end for**

22: $\mathbf{C} \leftarrow \text{GAT}(\mathbf{H}, \Theta_{GAT})$         $\triangleright$ Process all node features to get context vectors

23: $\mathbf{c}_i \leftarrow \mathbf{C}[i] \in \mathbb{R}^{128}$             $\triangleright$ Extract own context vector

24:                 $\triangleright$ — *Phase 4: Final Action Selection (Parallelized)* —

25: $\mathcal{A}_{\text{topK}} \leftarrow \{a \mid (a, e) \in \mathcal{E}_{\text{topK}}\}$

26: Initialize Q-values array $Q_{\text{vals}} \in \mathbb{R}^K$.

27: **for** $k = 1$ to $K$ **in parallel do**

28:      $a_k \leftarrow \mathcal{A}_{\text{topK}}[k]$

29:      $\mathbf{a}_k^{\text{enc}} \leftarrow \text{EncodeAction}(a_k) \in \mathbb{R}^7$

30:      $Q_{\text{vals}}[k] \leftarrow Q_i([\mathbf{s}_i || \mathbf{a}_k^{\text{enc}} || \mathbf{c}_i])$             $\triangleright$ Evaluate Q-value for action k

31: **end for**

32: $k^* \leftarrow \arg\max(Q_{\text{vals}})$

33: $a_i^* \leftarrow \mathcal{A}_{\text{topK}}[k^*]$

34: **return** $a_i^*$

---

buffer, where $\mathbf{S}_t = \{\mathbf{s}_1, \ldots, \mathbf{s}_N\}_t$ is the joint state, $\mathbf{A}_t$ is the joint action, and $R_t$ is the team reward.

2. **Team Reward Calculation:** The shared team reward $R_t$ is the actual, measured reduction in global cognitive free energy: $R_t = \mathcal{F}(t) - \mathcal{F}(t+1) = \Delta\text{EIG}_{\text{total}}(t+1)$. The global free energy $\mathcal{F}(t)$ is calculated by numerically integrating the information density functions over the entire known map volume, a computation that is only feasible in the centralized training environment.

3. **QMIX Training:** A batch of experience tuples is sampled from the replay buffer. For each tuple, the individual Q-networks produce local utility values $Q_i(\mathbf{s}_i, a_i, \mathbf{c}_i)$. These are fed into a monotonic mixing network, $Q_{\text{tot}} = f_{\text{mix}}(Q_1, \ldots, Q_N, \mathbf{S}_t)$, which estimates the total joint action-value. The mixing network and all individual networks are trained by minimizing the standard TD-loss:

$$\mathcal{L}_{\text{TD}} = \mathbb{E}\left[(y_t - Q_{\text{tot}}(\mathbf{S}_t, \mathbf{A}_t))^2\right] \tag{33}$$

where the target $y_t = R_t + \gamma \max_{\mathbf{A}_{t+1}} Q_{\text{tot}}(\mathbf{S}_{t+1}, \mathbf{A}_{t+1})$ is computed using a target network for stability. The gradients from this loss are backpropagated through the mixing network and down to the individual Q-networks and the GAT parameters.

This CTDE framework enables the policy to learn complex inter-agent dependencies and credit assignment, resulting in sophisticated emergent collaborative strategies that would be unattainable with purely decentralized learning.

### B.6 NETWORK ARCHITECTURES AND TRAINING HYPERPARAMETERS

To ensure the reproducibility of our results, this section provides a comprehensive specification of all neural network architectures, data flow tensor dimensions, and key training hyperparameters. All models were implemented in PyTorch and trained using the Adam optimizer.

#### B.6.1 DATA FLOW AND TENSOR DIMENSIONS

The collaborative decision-making process involves a precise flow of information between different network modules. The key tensor dimensions for a single agent are as follows:

- **Local State Embedding ($\mathbf{s}_i$):** The agent's local state is encoded into a **77-dimensional** vector. This vector is a concatenation of its 13-dim kinematic state (3D position, 4D quaternion orientation, 3D linear velocity, 3D angular velocity) and a 64-dim environmental feature vector, which is obtained by a lightweight CNN processing a local map patch generated from the neural field $\Phi_\theta$.
- **Action Intent Vector ($\mathcal{I}_i$):** Each agent selects its top-$K$ candidate actions to form an intent vector. We use $K = 5$. Each action is a 7D pose (3D position, 4D quaternion). The intent $\mathcal{I}_i$ is therefore a tensor of shape **5 x 8**, where each row contains the 7D pose and its corresponding scalar EIG value. This tensor is flattened to a 40-dim vector before being processed.
- **GAT Input Feature ($\mathbf{h}_j$):** The input feature for each node $j$ in the communication graph is created by concatenating its state embedding $\mathbf{s}_j$ (77-dim) and its flattened intent vector (40-dim). This combined 117-dim vector is then passed through an MLP encoder to produce a final feature vector $\mathbf{h}_j$ of dimension **128**.
- **Q-Network Input:** The Q-network evaluates a single action $a_j$ in the context of team intentions. Its input is a single flat vector of dimension **212**, formed by concatenating the agent's own state $\mathbf{s}_i$ (77-dim), the encoded action $a_j$ (7-dim pose), and the coordination context vector $\mathbf{c}_i$ (128-dim) received from the GAT.

#### B.6.2 ARCHITECTURAL AND TRAINING PARAMETERS

The detailed hyperparameters for each component are listed in Table 6.

The specific input to the **Q-Network** for an agent $i$ is a concatenation of its local state embedding $\mathbf{s}_i$ and the coordination context vector $\mathbf{c}_i$ produced by the GAT. The state embedding $\mathbf{s}_i$ itself comprises the robot's 13-dimensional kinematic state (pose, linear/angular velocities) and a 64-dimensional feature vector summarizing the local environment, which is generated by a lightweight convolutional encoder processing a local map patch. This detailed and standardized configuration is crucial for achieving the reported performance and serves as a solid foundation for future research building upon our framework.

## C ADDITIONAL EXPERIMENTAL RESULTS AND ANALYSIS

This section provides supplementary visualizations, detailed metric breakdowns, and discussions that were summarized in the main paper for brevity. These offer deeper insights into the framework's performance, behavior, and resilience.

### C.1 QUALITATIVE VISUALIZATIONS AND BEHAVIORAL ANALYSIS

The following visualizations offer deeper insights into the framework's internal processes and emergent behaviors, providing qualitative support for the quantitative results presented in the main paper.

Table 6: Architectural and training hyperparameters for the core components of the SCENE framework.

| Component | Parameter | Value |
|---|---|---|
| **1. Unified World Representation Fields ($\Phi_\theta$ and $\Psi_\xi$)** | | |
| **Hash Encoding** (Instant-NGP Backend) | Hash Table Levels ($L$) | 16 |
| | Feature Dimension per Level ($F$) | 4 |
| | Hash Table Size ($T$) | $2^{19}$ |
| | Coarsest Resolution ($N_{\min}$) | 16 |
| | Finest Resolution ($N_{\max}$) | 2048 |
| **Geometric Head MLP** | Network Structure | 3 layers $\times$ 128 width |
| | Activation Function | ReLU |
| **Semantic Head MLP** | Network Structure | 4 layers $\times$ 256 width |
| | Activation Function | SiLU (Sigmoid Linear Unit) |
| | Deep Ensemble Size ($N_e$) | 5 independent heads |
| **Dynamic Head MLP** | Network Structure | 2 layers $\times$ 128 width |
| | Activation Function | ReLU |
| **AoI Field MLP ($\Psi_\xi$)** | Network Structure | 2 layers $\times$ 128 width |
| | Activation Function | ReLU |
| **2. Collaborative Decision-Making Network** | | |
| **GAT** | Input Feature Dimension | 128 |
| | GAT Layers | 2 |
| | Attention Heads | 8 |
| | Output Feature Dimension (Context Vector $\mathbf{c}_i$) | 128 |
| **Q-Network** | Input Layer Dimension | 212 |
| | Network Structure | 3 hidden layers $\times$ 256 width |
| | Activation Function | ReLU (hidden), Linear (output) |
| **3. Training Hyperparameters** | | |
| **Neural Fields Training** ($\Phi_\theta, \Psi_\xi$) | Optimizer | Adam |
| | Learning Rate | $5 \times 10^{-4}$ (with exponential decay) |
| | Adam $\beta_1, \beta_2$ | (0.9, 0.999) |
| | Adam $\epsilon$ | $10^{-8}$ |
| | Batch Size (rays per agent) | 1024 |
| **MARL Training** (GAT & Q-Networks) | Optimizer | Adam |
| | Learning Rate | $3 \times 10^{-4}$ |
| | Adam $\beta_1, \beta_2$ | (0.9, 0.999) |
| | Adam $\epsilon$ | $10^{-8}$ |
| | Discount Factor ($\gamma$) | 0.99 |
| | Replay Buffer Size | $10^5$ transitions |

### C.1.1 Cognitive Evolution and Dynamic Perception

Figures 3 and 4 provide insights into the *process* through which SCENE achieves its superior performance. As shown in Fig. 3(a), SCENE's normalized cognitive free energy curve exhibits the steepest descent and converges to the lowest value. This is direct visual evidence of its efficient information acquisition, explaining the shorter Task Times in Table 1. The t-SNE visualization in Fig. 3(b) shows the system's understanding evolving through distinct "cognitive leaps" corresponding to significant discoveries.

This efficiency translates to superior dynamic perception. Fig. 4(a) shows SCENE's MOTA score quickly surpassing all others. The reason is revealed in Fig. 4(b), the spatio-temporal AoI heatmap. The regular vertical bands of low AoI are the signature of SCENE's intelligent revisit policy. Driven by the $EIG_D$ term, the system autonomously plans paths to "refresh" its knowledge of critical areas, ensuring its world model remains timely.

### C.1.2 Robustness, Scalability and Cognitive Divergence

The robustness of SCENE under network failure is visualized in Figure 5. The performance degradation surface shows that even under a severe combination of 50% communication loss and 25%

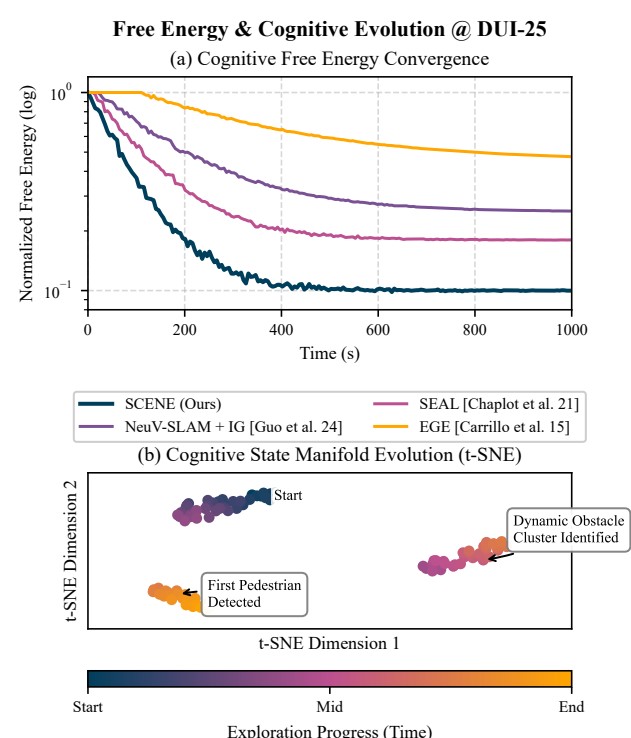

Figure 3: **Evolution of Cognitive State and Free Energy @ DUI-25.** This figure illustrates the efficiency of SCENE's information acquisition process. **(a) Cognitive Free Energy Convergence:** This plot shows the normalized global free energy (log scale) over time. SCENE's curve demonstrates the steepest descent and converges to the lowest value, providing direct evidence of its superior efficiency in reducing the team's overall uncertainty and novelty compared to baselines. **(b) Cognitive State Manifold Evolution (t-SNE):** This visualization projects the high-dimensional belief state of the world model into a 2D manifold. The trajectory, colored by mission progress, shows the system's understanding evolving from the initial state ('Start'). The distinct jumps and formation of new clusters (e.g., 'Dynamic Obstacle Cluster Identified') represent "cognitive leaps"—significant discoveries that fundamentally reshape the agent's world model.

robot dropout, SCENE maintains a high success rate and avoids catastrophic failure, unlike more brittle systems.

Figure 6 illustrates the trade-off between performance gain and system overhead. As team size grows, SCENE's task completion time (blue circles) decreases most rapidly. Critically, its coordination overhead (orange diamonds) grows slowly and linearly, demonstrating that our GNN-based intent-broadcasting mechanism achieves a near-optimal balance between performance and cost.

Finally, Figure 7 provides an intuitive understanding of why ablation models underperform. The *SCENE-NoSem* agent fails to discover the critical semantic target. The *SCENE-NoAoI* agent finds the target but fails to revisit a dynamic zone. The full SCENE agent is the only one to intelligently sequence these competing objectives, providing a clear, qualitative explanation for the superior performance metrics reported in Table 1.

## C.2 EVALUATION METRICS: DECOMPOSITION AND CONFIGURATION

To provide deeper insight into the comprehensive performance of SCENE, this section offers a detailed decomposition of the primary metrics presented in the main paper and the rationale for their configuration.

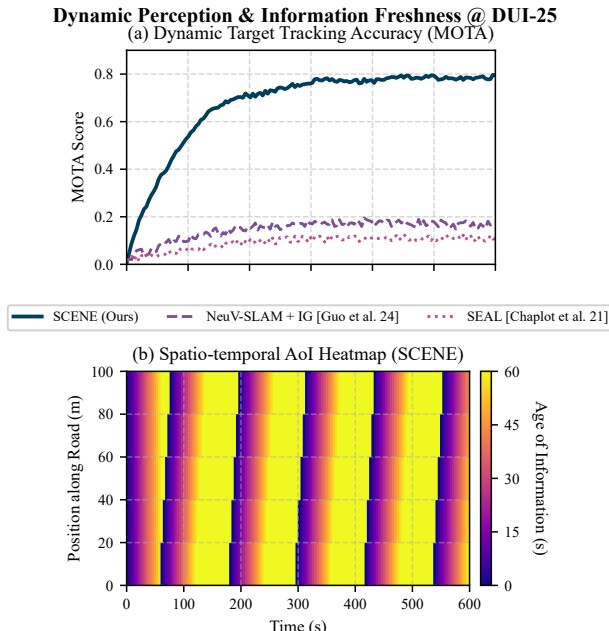

Figure 4: **Dynamic Perception Performance and Underlying Mechanism @ DUI-25.** This figure demonstrates SCENE's superior capability in dynamic environments and reveals the mechanism driving this performance. **(a) Dynamic Target Tracking Accuracy (MOTA):** The plot compares the Multi-Object Tracking Accuracy (MOTA) over time. SCENE rapidly achieves and sustains a high MOTA score, indicating its effectiveness in maintaining an accurate understanding of moving objects. In contrast, baselines not designed for dynamic scenes struggle to track targets. **(b) Spatio-temporal AoI Heatmap (SCENE):** This heatmap visualizes the Age of Information (AoI) for SCENE's world model across both space (y-axis, position along a road) and time (x-axis). The color represents the AoI, with warmer colors indicating staler information. The distinct vertical bands of cool colors (low AoI) are the signature of SCENE's intelligent revisit policy, driven by the $\mathcal{I}_D$ term in its objective function. This proactive "knowledge refreshing" behavior directly explains why SCENE maintains high tracking accuracy as shown in (a).

### C.2.1 COGNITIVE DISSONANCE (CD) METRIC DECOMPOSITION

The Cognitive Dissonance (CD) score, defined in Eq. equation 11, is a holistic measure of cognitive error. Table 7 decomposes the final CD scores from Table 1 into their constituent components for the challenging DUI-25 scenario. This allows for a granular analysis of each method's strengths and weaknesses across geometric, semantic, and dynamic domains. The results show that while a specialized method like NeuV-SLAM can achieve superior geometric accuracy, SCENE's balanced, unified approach allows it to achieve the best overall cognitive consistency by performing strongly across all three modalities simultaneously.

Table 7: Decomposition of the Cognitive Dissonance (CD) Metric in the DUI-25 Scenario. Lower values are better for all components. The final CD score is a weighted sum of these normalized error terms as defined in Appendix C.2.2.

| Method | Geometric Error ($\text{RMSE}_{\text{geom}}$) | Semantic Error ($1 - \text{mIoU}_{\text{sem}}$) | Dynamic Error ($1 - \text{MOTA}_{\text{dyn}}$) |
|---|---|---|---|
| **SCENE (Ours)** | 0.14 | **0.18** | **0.22** |
| NeuV-SLAM + IG | **0.12** | 0.65 | 0.75 |
| SEAL | 0.45 | 0.29 | 0.82 |
| LIMO + NBV | 0.21 | 0.80 | 0.85 |
| EGE | 0.65 | 0.85 | 0.59 |
| NBVP-FKIE | 0.82 | 0.90 | 0.68 |

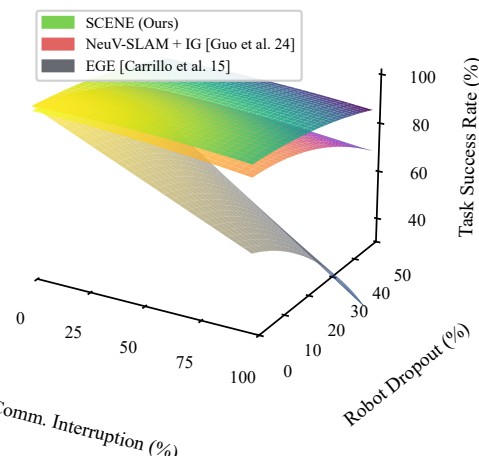

**Performance Degradation Surface**

Figure 5: **Performance Degradation Surface under Network Failure @ MARM-25.** This 3D surface plot visualizes the resilience of different methods by mapping Task Success Rate (z-axis) against increasing Communication Interruption (x-axis) and Robot Dropout (y-axis). The surface for SCENE remains consistently high and degrades gracefully, demonstrating exceptional robustness. In contrast, the surfaces for baselines like EGE show a steep decline, particularly with communication loss, highlighting their fragility in non-ideal network conditions. This illustrates SCENE's ability to maintain high performance even under severe operational stress.

### C.2.2 CONFIGURATION OF METRIC WEIGHTS

To ensure the objectivity and reproducibility of our experimental results, the weighting factors for our composite metrics, Cognitive Dissonance (CD) and Energy Efficiency (EE), were carefully chosen and held constant for all evaluated methods within a specific scenario. This appendix details the rationale and values used.

**Cognitive Dissonance (CD) Weights and Rationale**  The Cognitive Dissonance (CD) metric is designed as a holistic measure of the final cognitive error of the system. We introduce this unified metric because it moves beyond evaluating isolated aspects of performance (e.g., geometric accuracy alone) and instead assesses the total deviation of the agent team's final 'worldview' from ground truth. It quantifies the inconsistency across the three core cognitive dimensions—geometry, semantics, and dynamics—that our framework aims to unify. This approach better reflects a system's comprehensive cognitive capability, aligning the evaluation protocol with the unified theoretical foundation of our work.

The weights $\lambda_g, \lambda_s, \lambda_d$ in Equation equation 11 balance the contributions of these dimensions to the final CD score. The values were set to reflect the primary objectives of each benchmark scenario and were held constant for all evaluated methods to ensure a fair comparison:

- **Dynamic Urban Intersection (DUI-25):** This scenario emphasizes situational awareness in a dynamic environment. Therefore, we placed a higher emphasis on tracking accuracy to penalize methods that fail to maintain a timely understanding of moving agents. The weights were set to:
$$\{\lambda_g, \lambda_s, \lambda_d\} = \{0.3, 0.3, 0.4\}$$

- **Multi-Agent Rescue Mission (MARM-25):** This scenario is a task-driven mission focused on semantic search in a static, complex environment. As such, the dynamic component is not applicable ($\lambda_d = 0$). The weights were distributed to prioritize the successful and accurate identification of key semantic targets over pure geometric mapping. The weights were set to:
$$\{\lambda_g, \lambda_s\} = \{0.4, 0.6\}, \quad \lambda_d = 0$$

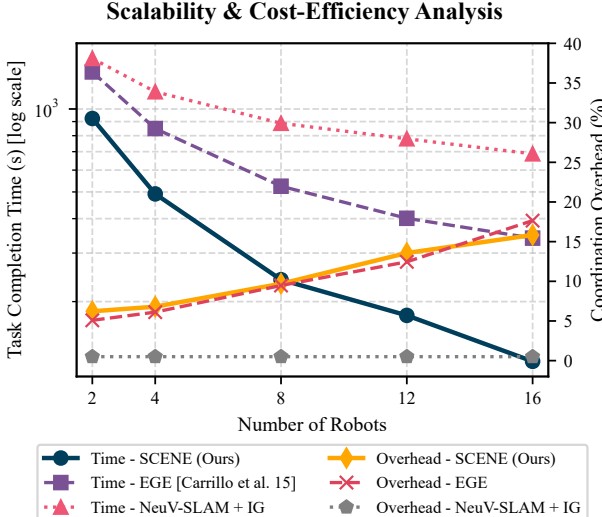

Figure 6: **Scalability and Cost-Efficiency Analysis.** This dual-axis plot evaluates performance scaling as the number of robots increases. The **left y-axis (log scale)** measures Task Completion Time, where lower is better. SCENE's performance (solid blue line) scales most effectively, showing a near-ideal reduction in mission time with more agents. The **right y-axis** measures Coordination Overhead, where lower is better. SCENE's GNN-based approach (solid orange line) exhibits a slow, linear growth in overhead, demonstrating its computational efficiency. This analysis reveals a key trade-off: while EGE also scales well in time, its overhead grows more rapidly. SCENE achieves the best balance, delivering superior performance scaling at a sustainable computational cost.

**Energy Efficiency (EE) Weight** The weight $\beta$ in Equation equation 12 quantifies the value of finding a semantic target relative to exploring free space. This is particularly relevant in the MARM-25 scenario. We set $\beta$ by estimating the average volume of a small room or corridor segment that a robot might have to explore to find a target.

- For the MARM-25 scenario, $\beta$ was set to **50 m³**. This implies that discovering one critical semantic target is considered as valuable as exploring 50 cubic meters of new space. This value was kept constant for all methods to provide a consistent measure of task-oriented efficiency.

By standardizing these weights, we ensure that our evaluation provides a fair and unbiased comparison of the different exploration strategies' ability to achieve a holistic understanding of their environment.

## C.3 QUANTITATIVE ANALYSIS OF HETEROGENEOUS STRATEGY

To provide concrete data supporting the emergent heterogeneous strategies discussed in the main paper (Figure 2 and Table 3), we quantified the distinct roles and contributions of Unmanned Ground Vehicles (UGVs) and Unmanned Aerial Vehicles (UAVs) during the MARM-25 mission. Table 8 breaks down key performance and behavioral metrics by agent type.

The data clearly shows that UAVs, leveraging their superior mobility and viewpoint, were responsible for the initial discovery of the vast majority of semantic targets (82%). However, their distant and often oblique views resulted in lower confirmation accuracy. UGVs, while slower in initial discovery, played a crucial role in navigating complex ground terrain to provide high-confidence confirmation (96% accuracy) of targets initially flagged by UAVs. This symbiotic relationship, evidenced by the high inter-class GNN attention reported in the main text, directly accounts for the heterogeneous team's superior overall performance.

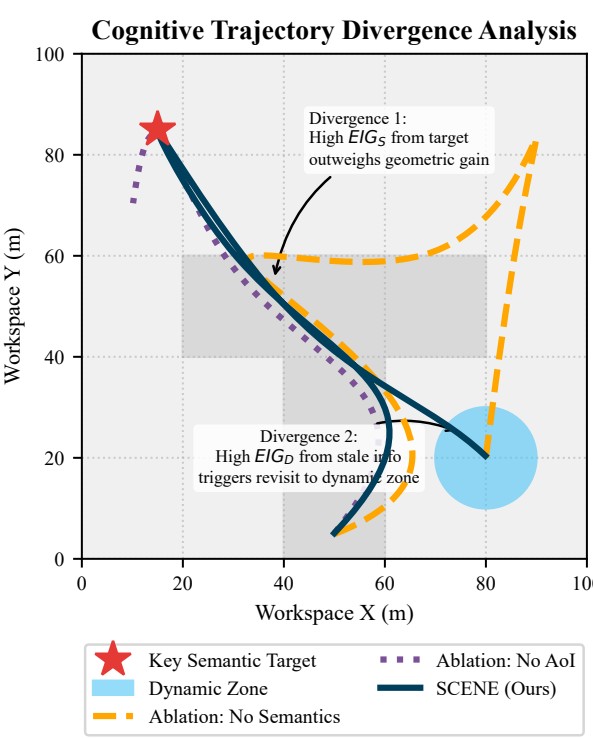

Figure 7: **Cognitive Trajectory Divergence Analysis.** This figure provides a qualitative explanation for the performance gaps observed in the ablation studies by visualizing the paths taken by different models. The mission space contains two key objectives: a high-value *Key Semantic Target* (red star) and a *Dynamic Zone* (blue circle) requiring periodic revisits. **SCENE (solid blue line):** Driven by its unified free-energy objective, it generates an optimal trajectory that successfully discovers the semantic target and then revisits the dynamic zone. **Ablation Models (dashed/dotted lines):** These models, lacking specific cognitive drives, produce sub-optimal paths. The *SCENE-NoSem* agent (orange dashes) completely ignores the high semantic gain ($EIG_S$) from the target. The *SCENE-NoAoI* agent (purple dots) finds the target but fails to act on the high dynamic gain ($EIG_D$) from the stale information in the dynamic zone. This clearly illustrates how the unified objective enables SCENE to intelligently sequence and service competing goals.

Table 8: Breakdown of Contributions by Agent Type in Heterogeneous Team (MARM-25).

| Metric | UAVs (2 Agents) | UGVs (6 Agents) |
|---|---|---|
| *Task Contribution* | | |
| Primary Targets Discovered (%) | 82% | 18% |
| Primary Targets Confirmed (%) | 25% | 75% |
| Avg. Confirmation Accuracy (%) | 68% | 96% |
| *Behavioral Profile* | | |
| Total Volume Explored (%) | 65% | 35% |
| Avg. Distance Traveled per Agent (km) | 2.1 km | 1.0 km |
| Avg. Time in Motion (%) | 78% | 62% |

## D    EXPERIMENTAL SETUP AND CONFIGURATION DETAILS

This section provides a comprehensive account of the experimental environment to ensure full transparency and reproducibility. We detail the construction of benchmark scenarios, the configuration of the simulation platform and robotic agents, and the implementation specifics of all baseline methods.

### D.1 HYPERPARAMETER SENSITIVITY ANALYSIS

The performance of the SCENE framework is modulated by the task-dependent weights $w_g, w_s, w_d$ in the EIG computation, which dictate the agent team's priorities. To assess the framework's robustness to variations in these weights, we conducted a sensitivity analysis in the MARM-25 scenario. In this static environment, $w_d$ is set to 0, and the trade-off is between geometric exploration ($w_g$) and semantic search ($w_s$), where $w_g + w_s = 1$. We varied the semantic weight $w_s$ from 0.2 to 0.8 and recorded the impact on key performance metrics. The results, averaged over 5 independent runs, are presented in Table 9.

The analysis shows that while the best performance is achieved around $w_s = 0.6$ (the setting used in our main experiments), the system does not exhibit catastrophic failure even when the weights are sub-optimal. The Task Completion Time remains within a reasonable range, degrading gracefully as the priority shifts away from the mission's primary semantic objective. For instance, even with a strong bias towards geometry ($w_s = 0.2$), the system still completes the mission, albeit much slower. This demonstrates that the SCENE framework is robust to the precise tuning of its objective weights, and its performance varies smoothly with changes in task priority, making it adaptable to different mission requirements without requiring exhaustive hyperparameter optimization.

Table 9: Sensitivity analysis of EIG weights in the MARM-25 scenario. The semantic weight $w_s$ is varied while keeping $w_d = 0$ and $w_g = 1 - w_s$. Performance is measured by Task Completion Time and Energy Efficiency (EE). The optimal setting used in the main paper is highlighted in bold.

| $w_g$ | $w_s$ | Task Time (s) $\downarrow$ | Energy Efficiency (EE) $\uparrow$ |
|-------|-------|---------------------------|-----------------------------------|
| 0.8 | 0.2 | 1580±75 | 0.35±0.11 |
| 0.6 | 0.4 | 810±40 | 0.68±0.09 |
| **0.4** | **0.6** | **450±20** | **0.92±0.07** |
| 0.2 | 0.8 | 520±25 | 0.81±0.08 |

### D.2 COMPUTATIONAL RESOURCE ANALYSIS

To provide a transparent assessment of the SCENE framework's computational demands, we present a detailed breakdown of resource utilization for our method and key learning-based baselines. The analysis, summarized in Table 10, was conducted on a simulated agent equipped with an NVIDIA RTX 4090 GPU, reflecting typical hardware for modern robotics research.

The results highlight the inherent trade-offs between different architectures. NeuV-SLAM is relatively efficient in memory due to its compact hash-grid representation but dedicates significant computation to high-frequency neural field queries for mapping. SEAL's overhead is dominated by its powerful 2D semantic segmentation backbone. SCENE, as a unified framework, exhibits a higher total resource footprint. Its memory usage stems from concurrently maintaining multiple neural fields and MARL-specific network components. The computational load is primarily driven by the online training and querying of these fields, which form the basis of its cognitive model.

Crucially, the additional overhead of SCENE directly translates into its advanced capabilities. For example, the 20% of GFLOPS dedicated to EIG computation and 10% to GAT collaboration are the explicit costs of intelligent, coordinated decision-making, which are absent in passive SLAM systems like NeuV-SLAM. This analysis confirms that SCENE's computational requirements are in a reasonable range for its complexity and are a direct investment in achieving a higher level of autonomous cognition.

### D.3 BENCHMARK SCENARIO CONSTRUCTION AND SIMULATION ENVIRONMENT

All experiments were conducted within **NVIDIA Isaac Sim 5.0** (August 2025 General Availability release), leveraging its high-fidelity physics and rendering capabilities. The simulation utilized the **PhysX 5.4** engine, with a physics step of 1/60 s, and the **RTX Real-Time** renderer for photorealistic sensor data generation.

#### D.3.1 COMMON ENVIRONMENT AND SENSOR CONFIGURATIONS

To ensure consistency, the following configurations were applied across all relevant scenarios:

Table 10: Detailed Computational Resource Breakdown. GFLOPS are reported per-agent, per-frame/decision-cycle. VRAM is the typical GPU memory footprint during operation. The GFLOPS breakdown shows the percentage of computation dedicated to each core module.

| Method | Total GFLOPS | VRAM (GB) | GFLOPS Breakdown (%) |
|---|---|---|---|
| **NeuV-SLAM + IG** | ~50-100 | 8-16 | Neural Field Queries: 50%
Map Rendering & Fusion: 30%
Pose Optimization: 20% |
| **SEAL** | ~100-200 | 6-12 | 2D Semantic Segmentation: 40%
Projection Mapping: 30%
Policy Network Inference: 20%
Other: 10% |
| **SCENE (Ours)** | ~200-400 | 10-16 | Neural Field Training/Queries ($\Phi_\theta, \Psi_\xi$): 50%
EIG Computation: 20%
GAT Collaboration: 10%
Q-Network Inference: 10%
Other (e.g., Prototype Memory): 10% |

- **Physics Materials:** Standardized physics materials were applied to all surfaces to ensure realistic interactions. For example, concrete surfaces in DUI-25 used 'Static Friction: 0.7', 'Dynamic Friction: 0.6', and 'Restitution: 0.2'. Rock and soil surfaces in MARM-25 and BCE-25 used 'Static Friction: 0.9', 'Dynamic Friction: 0.8', and 'Restitution: 0.3'.
- **Sensor Noise Models:** To simulate real-world sensor imperfections, physically-based noise was injected into sensor data streams via 'omni.sensors' graph nodes.
  - *RGB-D Camera:* A Gaussian noise model was applied to the depth channel with a standard deviation of 0.02 m.
  - *LiDAR:* A Gaussian noise model was applied to point cloud measurements with a standard deviation of 0.05 m.

### D.3.2    SCENARIO-SPECIFIC CONSTRUCTION DETAILS

Each benchmark scenario was custom-built to systematically test specific aspects of active cognition.

- **Dynamic Urban Intersection (DUI-25):** Designed to test long-term situational awareness in a large, dynamic world.
  - *Map and Rendering:* A 500m × 500m urban area featuring multiple intersections and buildings. The scene was rendered with RTX Global Illumination to produce realistic lighting and shadows.
  - *Dynamic Agents:* The scene was populated with 20 vehicles and 30 pedestrians. Agent behaviors were driven by a combination of 'ActionGraph' and Behavior Trees. Vehicles, controlled by a 'PhysX Vehicle Controller', followed road networks with speeds from 5-15 m/s, using collision detection nodes for local avoidance. Pedestrians, driven by 'Animation Graphs', executed random walks on sidewalks at 1-2 m/s, creating unpredictable encounters.
- **Multi-Agent Rescue Mission (MARM-25):** Designed to evaluate task-driven semantic search in a complex, unstructured 3D environment.
  - *Map Characteristics:* A 150m × 100m disaster site, comprising a partially collapsed three-story building and an outdoor debris field, featuring significant vertical complexity and non-planar surfaces.
  - *Semantic Targets:* The mission required locating 10 high-value semantic targets (rescue mannequins) and 5 secondary targets (fire extinguishers), often in locations with poor visibility.
- **Subterranean Challenge Environment (BCE-25):** Designed to test robustness in perceptually-degraded environments.
  - *Procedural Generation:* A 1.5 km long cave network was procedurally generated. The overall tunnel layout was defined by a segmented path, while wall geometry was sculpted using multiple octaves of Perlin noise with frequencies from 0.01 to 0.1 and amplitudes from 0.5 to 2.0 m to create realistic irregularities.
  - *Perceptual Challenges:* Environmental ambient light was set to a near-zero intensity ([0.01, 0.01, 0.01]) to enforce reliance on agent-mounted light sources. These were modeled as spot-lights with an intensity of $1 \times 10^6$ cd and a 45° cone angle. Wall textures were procedurally

blended using a 'Material Graph' and noise nodes to be sparse and low-contrast, challenging feature-based SLAM.

### D.4 SIMULATION PLATFORM AND AGENT CONFIGURATION

The heterogeneous robot team and their sensor suites were configured with realistic parameters, as detailed in Table 11.

Table 11: Robot Platform and Sensor Suite Configuration in Isaac Sim.

| Component | Parameter | Value |
|---|---|---|
| *1. Robot Platforms* | | |
| **Clearpath Jackal (UGV)** | Max Linear Velocity | 2.0 m/s |
| | Max Angular Velocity | 4.0 rad/s |
| **DJI Mavic (UAV)** | Max Horizontal Speed | 18.0 m/s |
| | Max Ascent/Descent Speed | 8.0 m/s / 6.0 m/s |
| *2. Sensor Suite (per agent)* | | |
| **RGB-D Camera** | Resolution | $1280 \times 720$ |
| | Horizontal FoV | 90° |
| | Depth Clipping Range | 0.1 m – 10.0 m |
| **LiDAR Sensor** | Scan Frequency | 10 Hz |
| | Points per Scan | 5000 |
| | Max Range | 100 m |
| | Horizontal FoV | 360° |
| *3. Communication Model* | | |
| **Team Communication** | Max Range | 100 m (line-of-sight) |
| | Bandwidth Limit | 10 MB/s |
| | Failure Model | Random packet drop (for robustness tests) |

## E  LLM USAGE STATEMENT

We utilized a large language model (LLM) solely for the purpose of refining grammar, punctuation, and phrasing in this manuscript. The LLM was not used for generating any of the core scientific content, such as the methodology, experiments, or conclusions presented herein.

