# OpenReview forum: "SCENE: Multi-Robot Active Cognition via Unified Free-Energy Minimization"
_ICLR.cc/2026/Conference — Submitted to ICLR 2026_

### Official Review · Reviewer_NBMG · 2025-10-27

**Soundness:** 3
**Presentation:** 3
**Contribution:** 4
**Rating:** 8
**Confidence:** 4

**Summary:**

This paper presents SCENE, a novel multi-robot framework that unifies geometric mapping, semantic understanding, and temporal awareness under the principle of cognitive free-energy minimization. SCENE formulates them as components of a single free-energy objective grounded in expected information gain. The framework integrates a multi-head neural implicit field (for geometry, semantics, and dynamics), a prototypical memory for open-world recognition, and a GNN for collaborative intent reasoning. Training is conducted end-to-end through multi-agent reinforcement learning, with the reduction of global free energy serving as a shared reward. Experiments in Isaac Sim demonstrate efficiency, robustness, and emergent collaboration.

**Strengths:**

This paper proposes an interesting reformulation of multi-robot exploration as cognitive free-energy minimization, enabling a principled balance between geometric uncertainty, semantic novelty, and information timeliness.
The introduction of a multi-head hash-encoded implicit field that continuously fuses geometry, semantics, and AoI dynamics is technically sound and well-motivated. The formulation of epistemic and novelty-based semantic gains through deep ensembles and prototypical memory is novel.
The design of an intent broadcasting + GAT-based coordination mechanism provides a scalable and interpretable solution for multi-agent collaboration, achieving (potentially heterogeneous/homogeneous) role differentiation between UAVs and UGVs.
The paper also presents experiments across diverse and dynamic environments with strong baselines. The visualization of GNN attention weights and unified utility maps offers rare interpretability into collaborative cognition.

**Weaknesses:**

*Conditional independence assumption may oversimplify multi-modal coupling*

Decomposition of expected information gain assumes conditional independence between geometry, semantics, and dynamics, which may not hold in complex sensory coupling (e.g., shape-color correlations). Theoretical justification or empirical sensitivity analysis is limited.

*Coordination, scalability, and communication overhead were not deeply evaluated*

Although intent broadcasting reduces bandwidth, the study does not examine scaling (beyond 16 robots). It remains unclear how GNN message passing or attention sparsity behaves under large team sizes or partial communication graphs.

*Limited analysis on real-world transferability*

All experiments are simulation-based. The practical challenges in real world are not yet validated, which slightly limits the claims of “autonomous cognition for multi-robot systems”.

**Questions:**

How robust is the unified free-energy objective when the independence assumption between geometric, semantic, and dynamic components breaks down? Could cross-correlation terms (e.g., mutual information between G and S) be incorporated?

How does the communication and computation scale beyond 16 robots? Has the team evaluated attention sparsification or hierarchical message aggregation (or a well-designed message format) for very large multi-robot teams?

To what extent does the learned collaboration depend on specific reward shaping? If the free-energy reduction reward were partially replaced with task-specific goals (e.g., exploration, object retrieval), would the emergent behavior persist?

---

### Official Review · Reviewer_AaGE · 2025-11-01

**Soundness:** 3
**Presentation:** 3
**Contribution:** 3
**Rating:** 4
**Confidence:** 4

**Summary:**

This paper introduces SCENE, a unified framework for multi-robot exploration that optimizes three objectives: geometric uncertainty reduction, semantic novelty discovery, and information timeliness (AoI). It does this by treating the problem as free-energy minimization. Experiments in Isaac Sim on three varied environments show SCENE outperforming several baselines, and its own ablations.

**Strengths:**

1. **Novel methodology.** The proposed framework is conceptually fresh—it unifies geometric, semantic, and temporal (AoI) objectives within a single free-energy–based optimization and integrates them through a shared neural implicit field and GNN-based MARL coordination.
2. **Comprehensive experiments.** The experimental section is thorough, covering multiple environments, detailed ablation studies, and robustness tests (e.g., communication loss and robot dropout). This extensive evaluation effectively supports the paper’s main claims.

**Weaknesses:**

1. **Intro and Related Work too compressed.** Motivation and prior work are blended into one short section, so it’s unclear what’s standard (NBV, semantic exploration, MARL coordination) vs. what’s new here. Please split and expand Related Work to clearly position against these baselines.
2. **Custom benchmarks and metrics reduce credibility.** The authors design three new benchmarks instead of using established ones, which weakens external validity. Similarly, their self-defined performance metrics (like Cognitive Dissonance and Energy Efficiency) lack clear grounding in prior literature.
3. **Inconsistent experimental setup.** Table 1 compares SCENE with baselines only on the DUI-25 environment, while the ablation study is conducted on MARM-25. This inconsistency makes it difficult to assess how general the method’s advantages are. Including baseline comparisons or ablations across multiple tasks would make the experimental evidence more convincing.

**Questions:**

1. What does “hash-encoded” mean in the phrase “hash-encoded, multi-head neural implicit field”?
2. Why do you choose QMIX instead of MADDPG or MAPPO?

---

### Official Review · Reviewer_3NE4 · 2025-11-07

**Soundness:** 2
**Presentation:** 1
**Contribution:** 2
**Rating:** 2
**Confidence:** 3

**Summary:**

The paper introduces SCENE, a unified framework for multi-robot active exploration and cognition, grounded in the principle of cognitive free-energy minimization. It formulates exploration as minimizing the discrepancy between internal world models and sensory observations, including geometric uncertainty, semantic novelty, and information timeliness. The framework uses a hash-encoded neural implicit field with three heads for geometry, semantics, and dynamics. To solve the collaboration decision-making problem, it uses Graph Neural Networks and is trained via multi-agent reinforcement learning. The experiments under Issac Sim show good performance of the proposed method.

**Strengths:**

1. The exploration formation is relatively novel. It bridges geometric, semantic, and dynamic factors under a single objective.

2. It unified world representation, cognitive state, and policy learning together in one framework, making full use of the powerful capabilities of existing models.

3. The system maintains high success rates even under communication loss and perceptual degradation.

**Weaknesses:**

**Presentation**

1. The overall presentation quality is weak. The paper contains many typographical errors, inconsistent reference formats, incorrect indentation, possibly irregular line spacing, and even the format of the title. These formatting issues significantly reduce readability and professionalism.

2. The paper lacks intuitive visual explanations. It mainly relies on dense text and formulas without providing a clear framework or pipeline diagram to help readers grasp the overall architecture. A method figure illustrating the interaction between the world model, semantic modules, GNN, and policy learning would be highly beneficial.

**Content**

3. Large portions of the appendix repeat content from the main text or from previous works (e.g., Appendix A). This redundancy dilutes the technical contribution and makes the paper unnecessarily long. The appendix should instead focus on additional experimental details or proofs.

4. The claimed novelty—a free-energy formulation integrating geometric uncertainty, semantic novelty, and Age of Information (AoI)—is not well supported in the main text. The proposed framework, SCENE, appears as a combination of existing components (semantic segmentation, implicit neural mapping, GNN coordination, MARL training) rather than a clearly novel formulation. Much of the main content describes prior methods rather than highlighting what is new. The manuscript would benefit from a sharper focus on how the proposed free-energy objective differs from or generalizes existing EIG-based exploration frameworks.

5. The reinforcement learning reward is defined purely as the reduction in global free energy, which essentially maximizes exploration. However, no task-specific objective is considered, making the learned behavior potentially suboptimal or misaligned with missions. The framework should clarify how goal-oriented collaboration emerges beyond exploration alone.

6. The paper merges Introduction and Related Work into one section, resulting in an incomplete discussion of prior literature. Given the heavy use of established methods, the authors should include a dedicated Related Work section addressing at least: Cognitive free-energy or active inference frameworks in robotics; GNN-based multi-robot coordination; Multi-agent reinforcement learning for exploration and collaboration, and so on.

**Experiments**

7. The experiments lack several important baselines and ablation studies, especially those testing the contribution of individual components in the free-energy formulation. Additionally, insufficient analysis of hyperparameters (e.g., weighting factors, only half a page with one simple result) is provided, leaving unclear how sensitive the performance is to these choices. More comprehensive evaluation and sensitivity analysis are necessary to substantiate the claimed robustness and generality.

8. All evaluations are conducted in simulated environments. Although results are promising, there is no validation on physical robots or real sensor data.

**Questions:**

1. In Appendix D.1, the authors discuss the trade-off between geometric exploration ($w_g$) and semantic search ($w_s$). How about $w_d$? Since it can not be set to 0 for all tasks.

2. How about comparing more recent large-scale multi-agent RL approaches beyond QMIX and SEAL, or just replacing RL with traditional planners? It lacks relevant ablation in reinforcement learning.

3. Where are these benchmarks derived? What are their objectives? Can you provide their sources?

4. Where does CD come from? There is no citation about the evaluation metric. How can you guarantee that this scoring is fair?

5. Given the popularity of LLM, could you provide some decision-making frameworks that use large models as agents, and compare them with your framework?

6. Please release the code and provide some robot demos, since it is a robot-related work.

---

### Meta-Review · Area_Chair_oRwp · 2025-12-22

**Summary:**

The main concern, shared by reviewers 3NE4 and AaGE (and alluded to by NBMG) is the positioning of the work relative to prior work on the free energy principle. As put by Reviewer 3NE4, "The manuscript would benefit from a sharper focus on how the proposed free-energy objective differs from or generalizes existing EIG-based exploration frameworks." Combined with additional concerns about experimental validity (e.g., custom metrics and environments) [3NE4, AaGE], I recommend that the paper be rejected.

**Reviewer Concerns:**

(see below)

**Reviewer Scores:**

Authors didn't post a rebuttal.

3NE4: 2
* "presentation quality is weak"
* "lacks intuitive visual explanations"
* "Large portions of the appendix repeat content from the main text or from previous works"
* "Much of the main content describes prior methods rather than highlighting what is new."
* "The framework should clarify how goal-oriented collaboration emerges beyond exploration alone."
* "incomplete discussion of prior literature ...: Cognitive free-energy or active inference frameworks in robotics; GNN-based multi-robot coordination; Multi-agent reinforcement learning for exploration and collaboration.":
* "experiments lack several important baselines and ablation studies"
* "insufficient analysis of hyperparameters"
* "no validation on physical robots or real sensor data"

AaGE: 4
* Intro and Related Work too compressed
* Custom benchmarks and metrics reduce credibility.
* Inconsistent experimental setup.

NBMG: 8
* Conditional independence assumption may oversimplify multi-modal coupling
* Coordination, scalability, and communication overhead were not deeply evaluated
* Limited analysis on real-world transferability

---

### Decision · Program_Chairs · 2026-01-26

Reject